# Turnkey photonic flywheel in a microresonator-filtered laser

Mingming Nie [1] ✉, Jonathan Musgrave [1], Kunpeng Jia [2] ✉, Jan Bartos[1], Shining Zhu [2], Zhenda Xie [2] ✉ & Shu-Wei Huang [1] ✉

Dissipative Kerr soliton (DKS) microcomb has emerged as an enabling technology that revolutionizes a wide range of applications in both basic science and technological innovation. Reliable turnkey operation with sub-optical-cycle and sub-femtosecond timing jitter is key to the success of many intriguing microcomb applications at the intersection of ultrafast optics and microwave electronics. Here we propose an approach and demonstrate the first turnkey Brillouin-DKS frequency comb to the best of our knowledge. Our microresonator-filtered laser design offers essential benefits, including phase insensitivity, self-healing capability, deterministic selection of the DKS state, and access to the ultralow noise comb state. The demonstrated turnkey Brillouin-DKS frequency comb achieves a fundamental comb linewidth of 100 mHz and DKS timing jitter of 1 femtosecond for averaging times up to 56 µs. The approach is universal and generalizable to various device platforms for user-friendly and field-deployable comb devices.

Dissipative Kerr soliton (DKS) frequency comb, generated by pumping an ultrahigh-quality-factor resonator, has been a ground-breaking technology with a remarkable breadth of demonstrated applications[1,2]. DKS frequency comb provides easy access to large comb spacing in nonconventional spectral ranges, enabling high-capacity communication with high spectral efficiency[3,4], ultrafast optical ranging with massive parallelism[5], and high-speed spectroscopy in the molecular fingerprinting region[6].

In time domain, DKS timing jitter is the key property that determines its applicability as an optical flywheel[7–9], where the pristine temporal periodicity with sub-optical-cycle timing jitter can be utilized for demanding applications at the intersection of ultrafast optics and microwave electronics, including photonic analog-to-digital converters (ADCs) for next-generation radar and communication systems[10–12], ultrafast sub-nanometer-precision displacement measurement for real-time probing optomechanics, ultrasonic phenomena, and cell-generated forces[13,14], coherent waveform synthesizers for pushing the frontiers of femtosecond and attosecond science[15–17], and timing distribution links for synchronizing large-scale scientific facilities like X-ray free electron lasers and the extreme light infrastructure[18–22].

Sub-optical-cycle and sub-femtosecond timing jitter have long been theoretically predicted to be the DKS timing jitter's quantum limit[23]. However, due to the excessive technical noises, such a quantum limit was not achieved until the two-step pumping scheme (see Supplementary Fig. S1 in Supplementary Information) was recently invented to mitigate the pump-to-comb noise conversion and fundamentally lower the DKS timing jitter towards the quantum limit[8]. The two-step pumping scheme is a universal principle that utilizes the Brillouin effect[8,9,24–26] and enables the free-running photonic flywheel demonstration in various platforms, including monolithic fiber Fabry-Pérot (FP) cavity[8,9], silica disk resonator[24] and silica wedge resonator[25]. The state-of-the-art Brillouin-DKS frequency comb demonstrated with the two-step pumping scheme achieves a fundamental comb linewidth of 400 mHz and DKS timing jitter of 1 femtosecond for averaging times up to 83 µs[9].

The remaining major obstacle to the widespread application of the Brillouin-DKS frequency comb is the lack of turnkey operation that

[1]Department of Electrical, Computer and Energy Engineering, University of Colorado Boulder, Boulder, Colorado 80309, USA. [2]National Laboratory of Solid State Microstructures, School of Electronic Science and Engineering, College of Engineering and Applied Sciences, School of Physics, and Collaborative Innovation Center of Advanced Microstructures, Nanjing University, 210093 Nanjing, China. ✉e-mail: mingming.nie@colorado.edu; jiakunpeng@nju.edu.cn; xiezhenda@nju.edu.cn; shuwei.huang@colorado.edu

mitigates the complex comb initiation dynamics and eliminates sophisticated feedback electronics (see Supplementary Fig. S1 in Supplementary Information) to achieve compact footprint, low power consumption, and environmental ruggedness for the long-sought-after goal of user-friendly and field-deployable comb devices.

Here, we devise and demonstrate the first turnkey Brillouin-DKS frequency comb, to the best of our knowledge, by combining the two-step pumping scheme and active laser gain that realizes decoupling between the pump generation and the comb generation. We build a microresonator-filtered laser[27–35] that consists of a comb-generating passive FP microresonator nested in a pump-generating active ring cavity (Fig. 1a). This configuration utilizing the pump-comb decoupling characteristic distinguishes our approach from the nonlinear self-injection locking (SIL) method implemented to put the conventional DKS frequency comb into the turnkey operation regime[36–38]. Unlike nonlinear SIL, our approach does not suffer from vulnerability to feedback phase fluctuation and enables the deterministic selection of DKS soliton numbers. In addition, we demonstrate the self-healing capability of returning to the original comb state from instantaneous perturbations. Recently, a self-emergent laser cavity soliton has also achieved phase insensitivity, self-healing capability, and deterministic selection of DKS soliton numbers[33]. More importantly, our approach allows access to the ultralow noise comb state. The turnkey Brillouin-DKS frequency comb achieves a fundamental comb linewidth of 100 mHz and DKS timing jitter of 1 femtosecond for averaging times up to 56 μs, rendering it suitable for photonic ADCs, ultrafast optical ranging, coherent waveform synthesizers, and timing distribution links.

## Results

Figure 1a shows the schematic of the microresonator-filtered laser that consists of a comb-generating passive FP microresonator nested in a pump-generating active ring cavity. The FP microresonator is made of graded-index multimode fiber (GRIN-MMF) (Fig. 1b), and its linewidth, quality (Q) factor, and free spectral range (FSR) are 695 kHz, $2.78 \times 10^8$, and 10.087 GHz, respectively (Fig. 1c, also see "Methods"). The SMF active ring cavity has an FSR of 26.8 MHz, much larger than the microresonator linewidth, and the embedded BPF has a bandwidth of

8.5 GHz, narrower than the microresonator FSR (see "Methods"). This arrangement ensures single-frequency laser oscillation in the active ring cavity, and such a single-frequency laser pump is coupled into the fundamental mode of the FP microresonator for intermodal excitation of cross-polarized stimulated Brillouin lasing (SBL, Fig. 1d, e) that in turn generates the DKS microcomb through the two-step pumping scheme[8,9]. The single-frequency laser pump and the cross-polarized SBL can be straightforwardly separated by a PBS, as shown in Fig. 1a, to achieve the required decoupling between the pump generation and the comb generation.

### Principle of turnkey Brillouin-DKS

In the two-step pumping scheme, as the pump frequency is swept from the blue-detuned side to near the resonance peak, SBL is first excited and then grows to generate Brillouin-DKS (see Supplementary Fig. S1 in Supplementary Information). If the offset frequency between the pump and Brillouin mode resonances is set to be slightly larger than the SBS frequency shift, the blue-detuned pump and red-detuned SBL can simultaneously exist in the microresonator. Their opposite thermal nonlinearities can compensate each out and render such Brillouin-DKS thermally stable and manually accessible with an expanded existence range. Here, the pump-generating active ring cavity keeps the pump detuning around the resonance peak (Fig. 2a) as the lasing will self-organize itself to the minimum loss and the maximum gain[39]. When the microresonator temperature (MRT) is set correctly such that the offset frequency is slightly smaller than the SBS frequency shift, the co-existing blue-detuned pump and red-detuned SBL in the microresonator manifests itself into the thermally stable DKS attractor. MRT is thus an effective control parameter that deterministically selects the DKS soliton number (see Fig. 3 and Supplementary Information Section II).

While the DKS attractor state is mainly defined by the microresonator, the turnkey dynamics closely follow the optical pathlength (OPL) change of the active ring cavity that is caused by refractive index increase and thermal expansion resulting from the EDFA pump absorption[40–42]. Figure 2b plots the pump frequency red shift during the laser turn-on process, which serves as a spontaneous

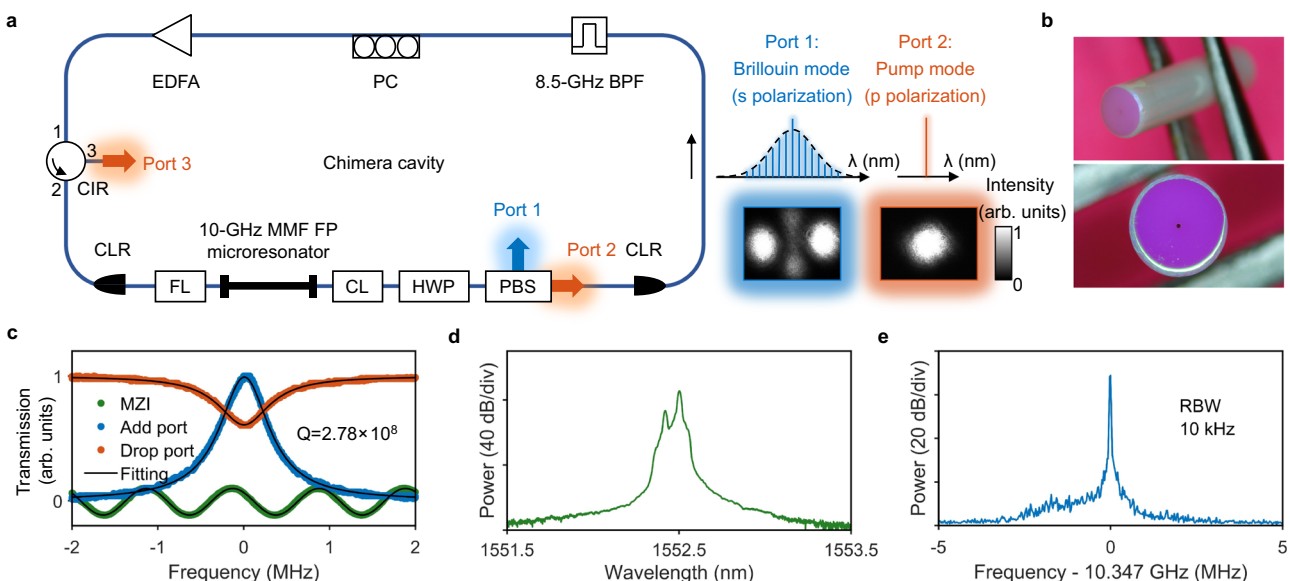

**Fig. 1 | Brillouin-DKS generation in a microresonator-filtered laser. a** Left: Schematic of the microresonator-filtered laser that consists of a ~10-GHz multi-mode fiber (MMF) FP microresonator nested in a single-mode fiber (SMF) ring laser cavity. EDFA Erbium-doped fiber amplifier, CIR circulator, CLR collimator, FL focusing lens, CL collimating lens, HWP half-wave plate, PBS polarization beam splitter, BPF band-pass filter, PC polarization controller. Right: spectra and beam profiles from PBS Port 1 and Port2. λ: wavelength. **b** Photographs of the MMF FP microresonator. **c** Frequency-calibrated transmission spectra of the pump mode showing the microresonator linewidth of 695 kHz. **d** Optical spectrum output from PBS Port 1, including the Brillouin mode and leaked weak-intensity pump mode. **e** Radio frequency (RF) beat note of the stimulated Brillouin scattering (SBS) frequency shift at 10.347 GHz.

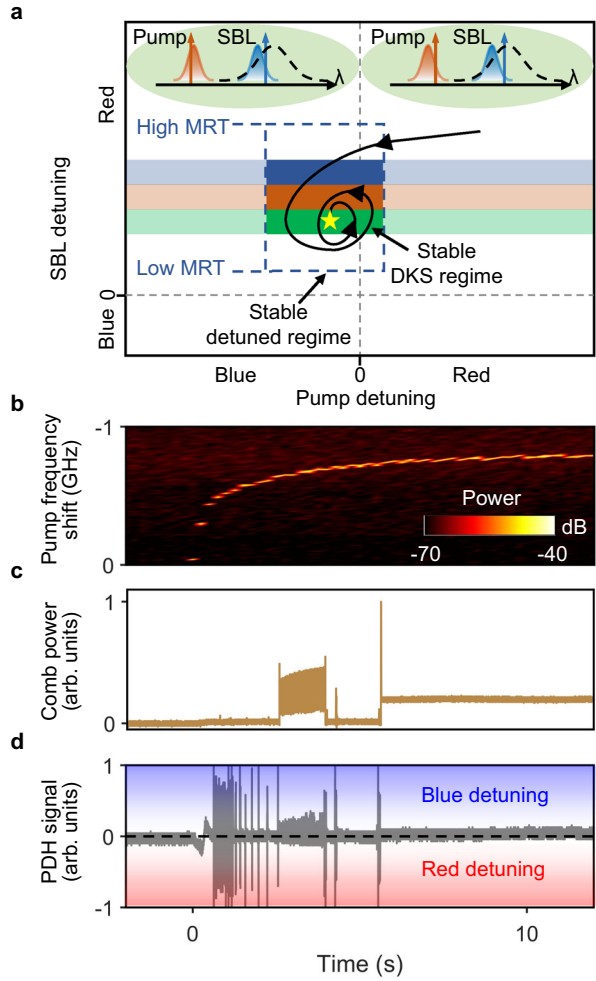

**Fig. 2 | Principle of turnkey Brillouin-DKS generation. a** The DKS attractor is defined by the intersection of thermally stable regime (dashed box) and SBL-detuning controlled DKS regime where the soliton number is color-coded. Of note, the microresonator temperature (MRT) changes the offset frequency between pump and Brillouin mode resonances, and consequently, MRT is an effective control parameter that determines the DKS attractor. Left inset: at the final stable DKS state (indicated by the yellow star), pump is blue detuned while SBL is red detuned. The black dashed line represents the Brillouin gain spectrum. Right inset: at an intermediate unstable state, both pump and SBL are red detuned. Evolution of **b** pump frequency shift, **c** comb power and **d** pump PDH signal during the turn-on process. PDH signal > 0: blue detuning, PDH signal < 0: red detuning, PDH signal = 0: zero detuning (resonance peak).

scanning of pump detuning to drive the system evolution into the DKS attractor state. Figure 2c, d shows the synchronously measured evolution dynamics of comb power and pump Pound–Drever–Hall (PDH) signal (see "Methods"), respectively. After several oscillations, stable DKS eventually forms when the pump laser is clamped at the slightly blue-detuned side of the microresonator resonance (Fig. 2d). We also numerically reproduce the turnkey soliton (see Supplemental Movies 1 and 2) and conduct comprehensive analysis in Supplemental Information Section III.

## Phase-independent turnkey Brillouin-DKS generation with deterministic soliton numbers

To demonstrate repeatable turnkey operation, the 980-nm laser of the EDFA is modulated by a chopper with a square-wave profile to mimic the turn-on process. As shown in Fig. 3a, soliton microcomb operation is reliably achieved, as confirmed by synchronously monitoring the comb power and the clean RF beat note of the comb repetition rate.

When the fiber cavity length is either fine-tuned with a resolution of 0.2 μm in a range of 10 μm or coarsely tuned with a resolution of 1 mm in a range of 20 mm, near 100% turnkey success probability can be achieved (Fig. 3b), indicating phase-independent and environment-insensitive turnkey operation, which is user-friendly and in sharp contrast to the nonlinear SIL method[36–38].

Figure 3c shows the optical spectra of a single-soliton state and perfect soliton crystal (PSC) states with two and three solitons, corresponding to repetition rates of 10.09 GHz, 20.18 GHz and 30.26 GHz, respectively. Besides PSC states, we did not observe any other multi-soliton states, such as soliton molecules or defects in the experiment. We attribute the dominant PSC existence over other multi-soliton states to the equally spaced potential well[43] created by co-lasing pump modes due to the insufficient BPF out-of-band suppression (see Supplementary Information Section IV). The RF beat notes of comb repetition rate with high contrast and single tone (right insets of Fig. 3c) indicate the stable mode-locking status. Of note, different soliton states are achieved via slightly changing the microresonator temperature by ~0.2 K thus changing the final stable SBL detuning when the system reaches thermal equilibrium. In addition, as shown in the left insets of Fig. 3c, the system can robustly evolve into the same soliton states with a probability of >90%, indicating a deterministic turnkey process.

## Strong immunity to perturbation

The turnkey Brillouin-DKS is strongly immune to environmental perturbation, including fiber cavity length change, vibration and temperature change. Figure 4a shows the soliton self-healing from instantaneous fiber cavity length change of 1 μm for the two-soliton state in Fig. 3c. Both the comb power and soliton repetition rate recover to the original state after the perturbation. Besides, the Brillouin-DKS can keep stable within a cavity length change of ±4 μm at a low evolving speed at 0.1 Hz, corresponding to a soliton repetition rate change of ±4.5 kHz (Fig. 4b). At the same time, the pump laser frequency shift is ±60 MHz while the pump detuning change is estimated to be ±100 kHz. Supplementary Figure S9 (see Supplementary Information Section V) also shows soliton self-healing from kicking the optical table and heating the fiber Bragg grating (FBG) based BPF by 1 K.

## Excellent noise performance

Figure 5a compares the single sideband (SSB) frequency noise spectra of the lasing pump mode, SBL and SBL comb lines, measured with a low-noise-floor optical frequency discriminator (see "Methods"). Thanks to the ultrahigh Q of the MMF FP microresonator, all the lasing pump mode, SBL and SBL comb yield a fundamental linewidth of ~100 mHz, representing the record-breaking narrowest DKS comb linewidth and approaching that of the state-of-the-art on-chip SBLs[44–46], but now at free running without any active electronics. In addition, the relative intensity noises (RINs) of the lasing pump mode, SBL and SBL combs are all below −120 dB/Hz above 1 kHz offset frequencies (Fig. 5b). Of note, 180-mW high power from Port 3 (Fig. 1a) is achieved for the single-frequency pump due to the non-critical coupling (Fig. 1c) and ~70% external coupling efficiency. The SBL combs, as well as the high-power lasing pump with low-frequency noise and low RIN, can benefit applications including coherent optical communications and optical atomic clocks. Owing to the narrow linewidth for both the lasing pump and the SBL, the achieved SBS frequency shift at 10.347 GHz is a good frequency synthesizer with low SSB phase noise of −107 dBc/Hz at 100 kHz (Fig. 5c), whose performance is comparable to one utilizing cascaded Brillouin process[46,47] but now self-starts without pumped by an external-cavity diode laser (ECDL). In Fig. 5c, we also plot the SSB phase noise of the SBL soliton repetition rate, measured with the all-fiber reference-free Michelson interferometer (ARMI) setup

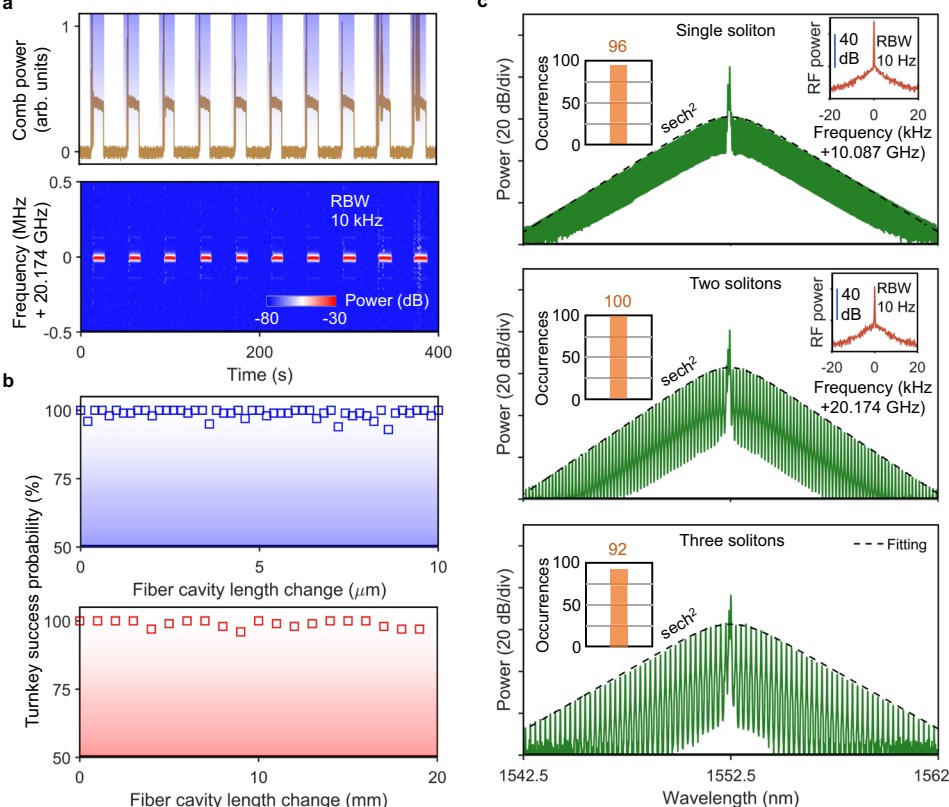

**Fig. 3 | Phase-independent turnkey Brillouin-DKS generation with deterministic soliton numbers. a** Ten consecutive switching tests. The top panel shows the measured evolution of comb power, while the bottom panel shows the measured electrical spectrum of the soliton repetition rate during the switching process. The EDFA is switched on periodically, as indicated by the shaded regions. **b** Turnkey success probability by changing fiber cavity length with fine-tuning (top)

and coarse-tuning (bottom). Each data point is acquired from 100 switch-on attempts. **c** Optical spectra of the turnkey SBL solitons. The black dashed lines show the fitted soliton spectral envelope. Left insets are the occurrences of a single soliton (top), two solitons (middle) and three solitons (bottom) among 100 times soliton generation. Right insets are the clean and high-contrast RF beat notes of comb repetition rate.

providing attosecond timing jitter resolution[8,9,48,49] beyond the capability of direct photodetection methods[50–52] (see "Methods"). The measured SSB phase noises at 10 kHz, 100 kHz, and 1 MHz offset frequencies are −128 dBc/Hz, −147 dBc/Hz, and −166 dBc/Hz, respectively. The timing jitter integrated from 18 kHz to 1 MHz is 1 fs (see Supplementary Fig. S10 in Supplementary Information Section VI), which is less than one-fifth of a single optical cycle at the SBL soliton center wavelength, reaching the photonic flywheel level. Compared with the soliton phase noise using nonlinear SIL method[38,53,54] and the phase noise of the SBS frequency shift (blue line in Fig. 5c), our Brillouin-DKS phase noise not only represents improvement of −10 dBc/Hz per decade, following $1/f^2$ trend with the offset frequencies, but also reaches lower noise of −166 dBc/Hz at 1 MHz offset frequency. The detailed noise analysis and performance comparison can be found in Sections VII to X of the Supplementary Information. The ultralow soliton phase noise can benefit critical applications, including photonic ADCs, ultrafast optical ranging, coherent waveform synthesizers and timing distribution links.

The long-term stability of the SBL soliton microcombs is also examined to be excellent in the laboratory environment (Fig. 5d). During an operating period of 2 h, the standard deviation of comb power and soliton repetition rate for the two-soliton state are 0.25% and 1.38 kHz with a resolution bandwidth of 1 kHz. The pump frequency shift and the pump detuning shift are also measured to be small as 200 MHz and 70 kHz, respectively, as shown in Supplementary Fig. S13 in Supplementary Information Section XI.

## Discussion

Our microresonator-filtered laser design, where pump generation in the active cavity and DKS generation in the microresonator are decoupled, is a universal topology for turnkey DKS generation and has the potential for fully on-chip integration. Silicon nitride Vernier microring filter has been proven effective as a narrow on-chip BPF[55]. Heterogeneously integrated semiconductor optical amplifier[37] and Erbium-doped silicon nitride waveguide amplifier[56] have both been recently demonstrated as viable on-chip gain media. SBL generation has been achieved in a weakly-confined silicon nitride microresonator[46] and a recent study further shows the potential of a tightly-confined silicon nitride waveguide[57]. Moreover, SBS is not the only intracavity effect that can be used to implement the two-step pumping scheme. Avoided mode crossing (AMX)[58–61] can also be utilized, and it relaxes the need to match the microresonator FSR with the SBS frequency shift, rendering AMX more flexible and user-friendly for the on-chip turnkey DKS microcomb implementation.

In conclusion, we demonstrate a universal approach to mitigate previous challenges and achieve reliable turnkey DKS frequency comb generation. Phase insensitivity, self-healing capability, deterministic selection of the DKS state, and access to the ultralow noise comb state are all successfully accomplished. The turnkey Brillouin-DKS frequency comb achieves a fundamental comb linewidth of 100 mHz and DKS timing jitter of 1 femtosecond for averaging times up to 56 μs, rendering it suitable for applications at the intersection of ultrafast optics and microwave electronics such as photonic ADCs, ultrafast optical ranging, coherent waveform synthesizers, and timing

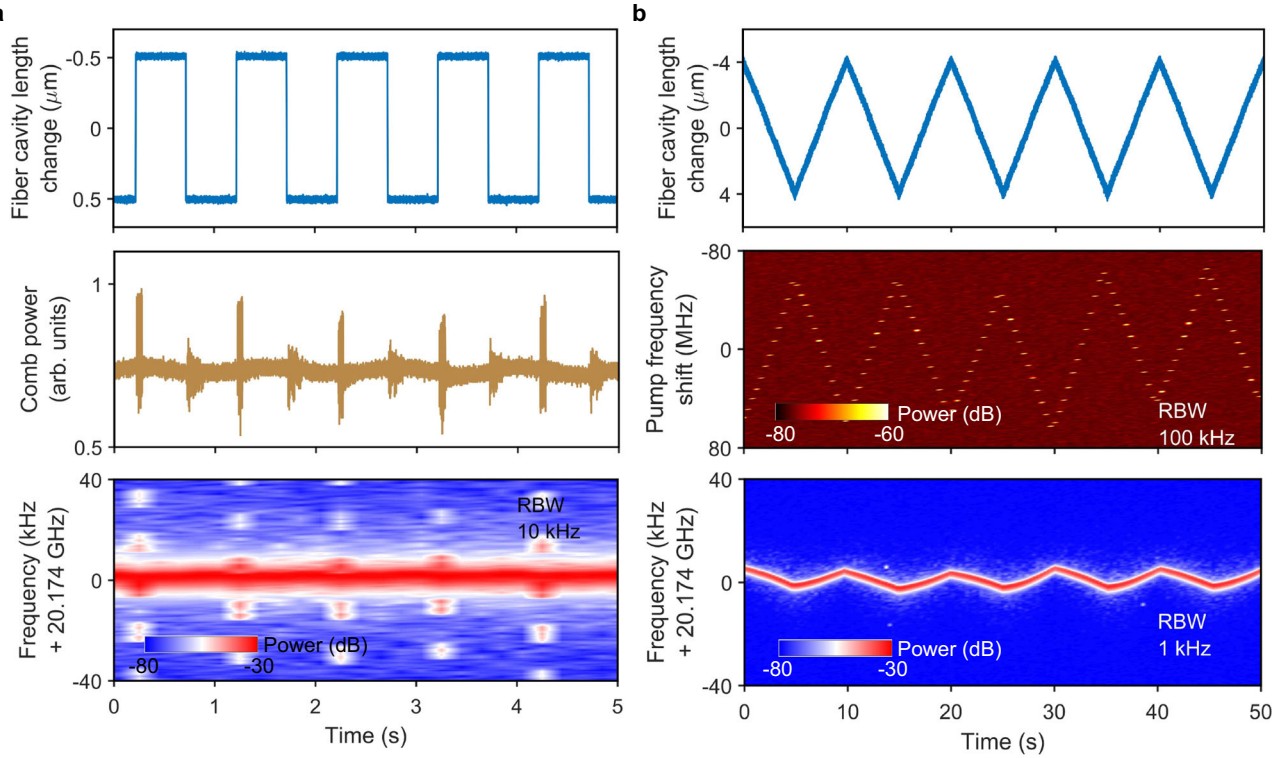

**Fig. 4 | Strong immunity to perturbation. a** Soliton self-healing from instantaneous perturbation to the fiber cavity length. **b** Soliton immunity to the slow modulation of the fiber cavity length.

distribution links. The approach can be generalized to photonic integrated circuit platforms for mass production of user-friendly and field-deployable comb devices.

## Methods

### MMF FP microresonator

Our high-Q MMF FP microresonator is fabricated through three steps: (1) commercial MMF (GIF50E, Thorlabs) is carefully cleaved and encapsuled in a ceramic fiber ferrule; (2) both fiber ends are mechanically polished to sub-wavelength smoothness; (3) both fiber ends are coated with optical dielectric Bragg mirror with reflectivity over 99.9% from 1530 to 1570 nm. The large-mode area of the MMF leads to low diffraction losses in the thick dielectric Bragg mirror coatings at both ends and results in ultrahigh Qs for both the internal pump mode and Brillouin mode[9].

The 10-mm long FP microresonator, corresponding to an OPL of ~29 mm, results in a cold-cavity free spectral range of 10.087 GHz, which is ready for microwave photonics applications. The group velocity dispersions of all the supported modes are simulated to be anomalous of ~−28 fs$^2$/mm[9].

The ultrahigh-Q microresonator has two roles here: (1) a high-finesse etalon filter in the fiber laser cavity responsible for single-frequency pump mode with low frequency noise; (2) the device generating SBL and corresponding DKS.

### Details of experimental setup

The fiber cavity includes 2-m gain fiber, 5-m passive fiber and 1-m freespace length, resulting in an OPL of 11.2 m and an FSR of 26.8 MHz. The home-made EDFA is bidirectional pumped, consisting of a 2-m highly Erbium-doped fiber (SM-ESF-7/125, Nufern), two 980/1550 nm wavelength demultiplexer (WDM) and two 980-nm diode lasers for each direction. The narrow BPF consists of a circulator and a temperature-controlled FBG centered at 1552.436 nm with a 3-dB bandwidth of 0.068 nm (8.5 GHz). In order to achieve low coupling

loss, free-space components are introduced to couple the light into and out from the MMF FP microresonator with a one-end coupling efficiency of ~70%. We believe all-fiber integration is feasible due to the compatibility between the MMF FP microresonator and other fiber components. The EDFA part and narrowband filter part with 3-m total bare fiber are shielded but not temperature-controlled, while the other components are exposed to the lab environment, including the 4-m passive fiber protected with a 0.9-mm jacket. The optical circulator ensures the unidirectional lasing of the fiber cavity and couples out the high-power single-frequency pump laser. The polarization controller is used to maximize the coupling efficiency into the microresonator.

The Q of the Brillouin mode is not measured due to the difficulty of finding the high-order MMF mode and the low coupling efficiency between the single-mode fiber and MMF. However, we can still estimate its Q to be larger than $1 \times 10^8$ according to our previous experiment data with the same kind of large-mode-area FP microresonators[9]. In addition, the total input pump mode power before coupling into the microresonator is as low as ~250 mW (at 980-nm total power of 1.5 W for the EDFA), also confirming the ultrahigh Q factor of the Brillouin mode. The MMF FP microresonator is temperature-controlled with a resolution of 10 mK (see Supplementary Information Section II).

A phase modulator is inserted in the active fiber ring cavity for all the experiments to monitor the pump detuning via the open-loop PDH error signal (see more details in Supplementary Information Section XII). The phase modulator is inserted before the BPF (Fig. 1a). The modulation frequency is 1 MHz and the PDH signal is demodulated by the single-frequency pump laser output from the circulator. The low-pass filter used in the PDH signal demodulation is 100 kHz. The modulation voltage of the phase modulator is chosen to be low without perturbing the SBL soliton generation and turnkey operation. Since the PDH error signal measures the overall pump detuning, any resonance frequency shift induced by thermal and nonlinear effects will be captured in the PDH error signal.

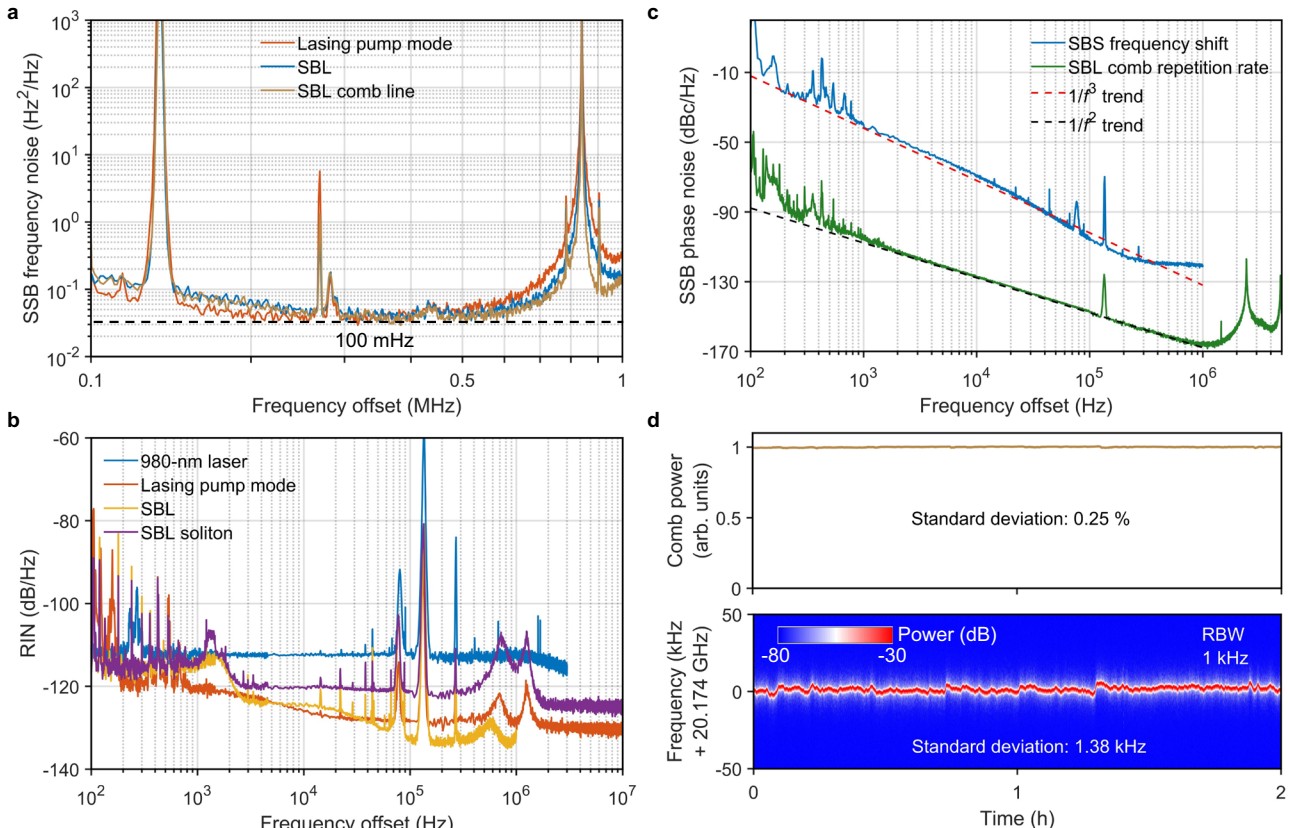

**Fig. 5 | Excellent noise performance. a** SSB frequency noise spectra of the lasing pump mode, SBL and SBL comb line. Fundamental linewidths are calculated from the white noise floor of the measured SSB frequency noise spectra. The same fundamental linewidths of both pump laser and SBL and weak linewidth narrowing factor from the SBS process are attributed to the limit of laser RIN (see Supplementary Information Section VIII). **b** RIN of 980-nm laser, pump mode, SBL and SBL soliton. **c** SSB phase noise spectra of the SBS frequency shift (10.347 GHz) and repetition rate of single SBL soliton (10.087 GHz). The coherent artifacts at 2.5 MHz and its harmonics (green line) result from the 82-m delay fiber used in the ARMI setup, while the peak at 135 kHz comes from the EDFA pump RIN. **d** Long-term stability of the comb power and comb repetition rate over 2 h in the laboratory environment with temperature variation of ±1 K.

The pump laser frequency shift is monitored by the beat note between the pump laser and a tunable ECDL that has a frequency stability of 2 MHz during the measurement. The beat note is measured by an electrical spectrum analyzer (E4407B, Keysight) at 4 Hz to show its evolution.

The comb power is monitored by a filter centered at 1554.5 nm with a 3-dB bandwidth of 0.5 nm.

The fine-tuning of the fiber cavity length at the micrometer level is realized by a piezoelectric stack, whose voltage-displacement curve is calibrated by an MZI.

We choose the 2-FSR perfect soliton crystal to characterize the turnkey soliton performance in Figs. 3 and 4 due to three reasons: (1) It is easy to tell the soliton energy source comes from the SBL instead of the pump laser according to the optical spectrum analyzer with a resolution of 0.02 nm. However, it is difficult to tell whether the DKS's pump is from the pump or the SBL at the single-soliton state. (2) The 2-FSR perfect soliton crystal state can convincingly demonstrate the soliton generation with a deterministic state for each turnkey operation when the system parameters are correctly set; (3) The 2-FSR perfect soliton crystal has a repetition rate of 20.18 GHz, which can still be measured and processed by our RF electronics.

### Measurement of laser phase noise and fundamental linewidth

A self-heterodyne frequency discriminator using a fiber-based unbalanced MZI and a balanced photodetector (BPD)[9] is employed to measure the laser phase noise and fundamental linewidth. One arm of the unbalanced MZI is made of 250-m-long single-mode fiber, while the other arm

consists of an acousto-optic frequency shifter with a frequency shift of 200 MHz and a polarization controller for high-voltage output. The FSR of the unbalanced MZI is 0.85 MHz. The two 50:50 outputs of the unbalanced MZI are connected to a BPD (PDB570C, Thorlabs) with a bandwidth of 400 MHz to reduce the impact of detector intensity fluctuations. The balanced output is then analyzed by a phase noise analyzer (NTS-1000A, RDL). The minimum fundamental linewidth that can be measured by this frequency discriminator is below 10 mHz.

### Measurement of comb repetition rate phase noise and timing jitter

The RF beat note of the soliton repetition rate is directly detected by a fast photodetector (EOT, ET-3500F) and measured by an electrical spectrum analyzer (E4407B, Keysight) at 5 Hz to show its evolution. Since the soliton phase noise measurement based on fast photodetectors is limited not only by the shot noise but also the available electronics operating at high frequency, we introduce the ARMI setup[8,9,48,49] to precisely measure the phase noise of SBL soliton. Two spectral regions of 1548.5 nm ± 0.25 nm (3 dB) and 1556.5 nm ± 0.25 nm (3 dB) are filtered out and sent to the interferometer. The locking bandwidth of the ARMI setup[9] is set to be 100 Hz. Therefore, the phase noise spectrum outside the locking bandwidth above 100 Hz is measured as shown in Fig. 5c.

We electrically divide the RF beat note of the SBS frequency shift of 10.347 GHz by 8 times to 1.29 GHz and then measure its phase noise with a downconverter (DCR−2500A, RDL) and a phase noise analyzer (NTS-1000A, RDL).

## Reporting summary

Further information on research design is available in the Nature Portfolio Reporting Summary linked to this article.

## Data availability

The data that support the plots within this paper is available on Zenodo (https://zenodo.org/records/10199122). All other data used in this study are available from the corresponding author upon request.

## Code availability

The analysis codes will be available on request.

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

## Acknowledgements

We thank Professor S. A. Diddams from NIST, Boulder, for fruitful discussions. We thank Dr. Bowen Li for the fruitful discussion on the manuscript revision. M.N., J.B., and S.W.H. acknowledge the support from the National Science Foundation (ECCS2048202), Office of Naval Research (N00014-22-1-2224), and National Institute of Biomedical Imaging and Bioengineering (REB029541A). J.M. is supported by the Department of Defense (DoD) through the National Defense Science & Engineering Graduate (NDSEG) Fellowship Program. K.J., S.Z., and Z.X. acknowledge the support by National Key R&D Program of China (2022YFA1205100, 2019YFA0705000, 2023YFB2805700), National Natural Science Foundation of China (62293523), Leading-edge technology Program of Jiangsu Natural Science Foundation (BK20192001), Zhangjiang Laboratory (ZJSP21A001), Guangdong Major Project of Basic and Applied Basic Research (2020B0301030009), Program of Jiangsu Natural Science Foundation (BK20230770).

## Author contributions

M.N. and S.W.H. conceived the idea of the experiment. M.N., J.B., and S.W.H. designed and performed the experiment, J.M. and M.N. performed the numerical simulation, and K.J., S.Z., and Z.X. designed and fabricated the microresonator. M.N. and S.W.H. conducted the data analysis and wrote the manuscript. All authors contributed to the discussion and revision of the manuscript.

## Competing interests

M.N. and S.W.H. are the inventors of a provisional patent application (U.S. Application No. 63/074665) filed by the University of Colorado Boulder on the turnkey frequency comb generation approach. The remaining authors declare no competing interests.
