## [Peer Review File · Nature Communications]

REVIEWER COMMENTS

Reviewer #2 (Remarks to the Author):

In their manuscript titled "Turnkey photonic flywheel in a Chimera cavity," the authors proposed and demonstrated the turnkey operation of Brillouin-Kerr soliton frequency combs using a Chimera cavity configuration, which consists of an FP microresonator nested in a fiber ring cavity. The concept is both fascinating and novel, as it breaks the traditional paradigm that the pump and the comb are co-generated in the same cavity in Brillouin-Kerr frequency combs. The decoupling scheme, along with the two-step pumping method, enables reliable turnkey operation, which is highly valuable for practical applications. Besides, the high coherence of the Brillouin-Kerr frequency combs demonstrated here can benefit areas such as ultrafast optics and microwave electronics.

Overall, the manuscript is well-written and detailed, and the experimental demonstrations are of exceptional quality. In my opinion, this work aligns well with the scope of Nature Communications, given its novelty and potential for broad interest. With a few minor modifications, I am convinced that this paper is suitable for publication in the journal.

The following are some places that may need further clarification:

1. I noticed that the authors refer to the Brillouin-DKS frequency comb with sub-optical-cycle timing jitters as a "photonic flywheel." However, I found that apart from the authors' previous work and ref. [7], there are few other papers that use this term. I would request the authors explain why the pristine temporal periodicity with sub-opticalcycle timing jitter is referred to as a photonic flywheel.
2. The authors state that "From the quantum noise perspective, fiber cavity is a superior platform as its weaker mode confinement and lower Kerr nonlinearity preferably minimize the quantum limit towards sub-femtosecond DKS timing jitter in microsecond time scale." However, in a recent publication in Optics Letters, Vol. 46, Issue 7, pp. 1772-1775 (2021), quantum-limited DKS timing jitter was also achieved in a WGM cavity with a similar level of timing jitter to that of the present work. Could the authors comment on this?
3. The paper provides a rather brief explanation of the turnkey principle. Could the authors provide the derivation of the attractor curve, the thermally stable regime (dashed box), and the boundaries of the various soliton states in Fig. 2a?
4. The authors mentioned that the MRT is an effective control parameter that deterministically select the DKS soliton number. Could the authors provide a specific derivation for this?
5. How the repetition rate frequency evolution in Fig. 3a was measured?
6. It seems that the schematic in Fig. 1a is missing the 980 nm pump laser.
7. In Fig. 5a, the linewidths of all the lasers are significantly lower than the corresponding linewidths in the WGM microresonator [Phys. Rev. Lett. 126, 063901 (2021)]. Could the authors provide a quantitative analysis of this difference?

8. In Fig. 5c, the SBS frequency shift follows a $1/f^3$ trend, while the SBS comb repetition rate follows a $1/f^2$ trend, and the noise level of the SBS comb repetition rate is much lower than that of the SBS frequency shift. Could the authors provide further analysis of these results?

Reviewer #3 (Remarks to the Author):

The paper 'Turnkey photonic flywheel in a Chimera cavity' by Nie and co-workers deals with the self-starting of dissipative Kerr solitons (DKS) in a microcavity filtered fibre laser.

The paper leverages the Brillouin effect and I find that it provides an important capability for the field. The authors, together with other groups (e.g. [23]), are establishing important results in the use of the Brillouin effect in microresonators. In this regard, the recent work of Baicheng Yao and coworkers, Phys. Rev. Lett. 130, 153802, should probably be cited too.

I find also very interesting the use of the Brillouin effect in a microcavity-filtered configuration. The authors explored a microcavity-filtered configuration already in their recent work (Light: Science & Applications volume 11, Article number: 296 (2022)). Differently from this, here the authors use an intracavity narrowband filter to isolate only a continuous wave that lases in the whole system sustaining the soliton in the Kerr cavity.

I have to say, however, that the authors appear to ignore a good chunk of literature on microresonator-filtered lasers. These approaches have been largely investigated for more than 10 years, see Nature Communications 3, 765 (2012), Nature Photonics 13 (6), 384-389 (2018). The configuration using a narrow filter to allow only a CW pump to oscillate was indeed proposed in 2013, Optics Express 21 (11), 13333-13341, and largely used for quantum microcomb generation, e.g. Nature Communications 6 (1), 8236. The authors call the microcavity-filtered approach a 'Chimera cavity'. I find that this confuses the literature. 'Chimera' is a well-known term in the nonlinear laser literature to describe a specific type of state (chaotic + stable state, see, e.g. 'Laser chimeras as a paradigm for multistable patterns in complex systems' Nature Communications volume 6, Article number: 7752 (2015)), completely out of context here. I would recommend keeping using the established terminology and changing the title to 'Turnkey solitons in a microresonator-filtered laser via Brillouin effect'.

In their introduction, the authors compare their approach only to self-injection locking (SIL). They state:

'Unlike nonlinear SIL, our approach does not suffer from the vulnerability to feedback phase fluctuation and enables the deterministic selection of DKS soliton numbers. In addition, we demonstrate the self-healing capability of returning to the original comb state from instantaneous perturbations. More importantly, our approach allows the access to the ultralow noise comb state'.

These properties have all been demonstrated already with microresonator-filtered microcombs, in laser-cavity soliton configuration. This should be properly acknowledged. Conversely, well after their introduction, they note:

'Of note, the Chimera cavity shares some similarity with the self-emergent laser cavity-soliton (LCS) [30] but the working principles are distinctly different. Self-emergent LCS critically depends on the parameters of the outer laser cavity which is more susceptible to environmental disturbance. Ultralow comb noise and long-term stability have not been demonstrated with the self-emergent LCS approach.'

While I am not convinced that the principles are distinctly different, it is surely not correct that 'Ultralow comb noise and long-term stability have not been demonstrated'. Beyond Ref [30], where the system was shown to operate on the hour timescale, the metrological properties have been more recently discussed in Applied Physics Letters 122 (12), 121104, showing both low noise generation, long term, state of the art free running stability. It is incorrect that 'Self-emergent LCS critically depends on the parameters of the outer laser cavity which is more susceptible to environmental disturbance,' Applied Physics Letters 122 (12), 121104 shows that the external parameters can be largely modulated while retaining the solitary generation and stability properties. The authors should, hence, avoid justifying the novelty of their work with unfactual statements on demonstrated results and instead provide an interpretation of their data that highlights the critical differences and possible opportunities.

Regarding the distinction between the working mechanism between this paper and the LCS, I think that there are many similarities and specific differences, which make the paper relevant. This requires, however, a fair comparison with what has been already demonstrated.

To me, this paper is an alternative pathway to control the slow mechanism that can induce soliton microcombs self-emergence in two cavities system, as demonstrated in [30]. Although the Brillouin effect has been explored already with externally pumped sources, with the mechanism that the authors summarise in their extended Fig. 1 and previously, e.g. in [23], its use in a microresonator filtered configuration is not obvious and merit publication. Before being ready for it, there are however some points that need to be clarified, together with a proper amendment and a fair discussion of the literature.

In terms of slow-fast nonlinear interaction, the authors use a similar principle of self-emergent LCS. They force a soliton state to be the dominant attractor of the system and, to this aim, they act on a global parameter to balance the nonlinearities of the system. This is the physical concept developed in [30], where the global parameters were cavity loss and pump energy. Here, specifically, they use the temperature: 'When the microresonator temperature (MRT) is set correctly such that the offset frequency is slightly larger than the SBS frequency shift, the co-existing blue-detuned pump and red-detuned SBL in the microresonator manifest itself into the thermally stable DKS attractor.'

This is the same principle used in [30], but with an interesting twist, because the Brillouin effect was not present in [30]. The Brillouin shift works to compensate for the thermal nonlinearity of the microcavity and brings the system to work on the red-detuned slope of the Kerr resonator, inducing the solitons. I note, however, that this was already discussed in [23] and with the two-step pumping in [8,9]. The authors report improved robustness by using the microcavity nested within the amplifying cavity. Hence, I would suppose that the amplifying cavity plays a role here.

Indeed, in [30] the balance of the thermal nonlinearity of the Kerr microcavity was completely obtained with the nonlinearity of the laser cavity only. It remains the question then, of how much the external

cavity plays a role in providing a more robust locking of the state also in the present work. Otherwise, it is unclear why the Brillouin cavity needs to be inserted in the amplifying loop and cannot simply be externally connected to the pump laser as in the Extended Fig. 1 and previous works.

Interestingly, the authors state that they observe a balance between the nonlinear shift induced by the Brillouin effect in the microcavity and the thermal effects in the erbium fibre. The authors comment

‘While the DKS attractor state is mainly defined by the microresonator, the turnkey dynamics closely follows the optical pathlength change of the active ring cavity that is caused by refractive index increase and thermal expansion resulting from the EDFA pump absorption [32–34].’

The authors follow the same citations [32–34] discussed in [30] to highlight this phenomenon. In case of the LCS, it has been demonstrated that this effect balances the nonlinearity in the microcavity to induce the soliton. The authors should then quantify ‘the optical pathlength change of the active ring cavity’ against the Brillouin shift and thermal microcavity nonlinearity to provide a proper physical insight in the principle, otherwise, there is no evolution from previous work on Brillouin states, e.g. [23], nor a distinct novelty compared to [30].

The MRT temperature is the external parameter that allows the system to achieve the red-locking microcomb states, which is the key for the soliton generation. In the supplementary, they state that soliton generation occurs in a range of 0.8 K. This is a narrow range of temperature, which I would expect to be susceptible not only from environmental changes (what temperature stabilisation accuracy are they using on the microcavity?), but also from power changes within the system. Can the authors report the effective working temperature of the microcavity? Is it always the same? Can the author comment on the dependence of this range with respect to the circulating power within the laser? Do the authors observe a dependency on the losses of the system or on the power balance between the two lasing polarisation components?

The authors comment ‘Of note, our deterministic turnkey process is independent of 980-nm laser power of the EDFA in an offset range of ± 100 mW’. This number makes little sense if it is not related to the actual gain and average power circulating in the laser. Can the author please comment on the change of laser power that they obtain within the ± 100 mW change of their pump power? How much is the change of temperature in the microcavity and its resonance shift?

On a minor point, I find the description of the Pound–Drever–Hall (PDH) modulation not clear. An additional figure to show these elements within the cavity would be helpful. The authors refer to a PDH signal and describe a MHz modulation of the intracavity pump. Usually, a PDH approach includes also an electronic feedback mechanism to lock the laser, e.g. on a filter slope or another laser. Here, however, such locking is not mentioned and seems not present. It appears that the authors only modulate the signal and use the MHz beating to track its changes against an ECDL reference laser. If it is so, calling this approach PDH is confusing. Could the authors comment on this and discuss if any other feedback mechanism against the laser power is used? The authors state ‘Thermal effect and Kerr effect are considered in the PDH signal measurement’. Can they please clarify how they take them into account in the measurement? Can the author comment on the stability of the ECDL laser used as the reference for this measurement?

Regarding the capability of deterministically selecting the DKS soliton number, I believe that the authors do not explain clearly why they obtain them. It should be investigated further. This particular aspect is different from the LCS. In [30], the deterministic generation of a certain soliton number occurs because the lasing light is also pulsed, and the self-locking mechanism is energy dependent. This allows the discrimination of the soliton number. Here however the lasing pump is CW, like in a Lugiato-Lefever or a self-injection locking system. In these approaches, the soliton number, and also the relative position of the solitons within the crystal, are usually dependent on additional phase parameters, like in ref [35]. Considering that the authors obtain a two-soliton state with always the same phase, I would agree with the authors that this appears to be similar to [35], as they comment:

‘The deterministic perfect soliton crystal states are believed to be caused by either the pump-SBL pair induced cross-phase modulation [35] or two co-lasing lasers due to the insufficient filtering effect of the BPF during the turnkey process’.

It must be stated, however, that this is a phase phenomenon (‘pump-SBL pair induced cross-phase modulation’). I would guess that it will depend on the relative frequency between the pair. Can the authors comment on that and on their ability to control different types of states? Do they have a specific range of parameters where they can achieve deterministic generation?

In conclusion, this is an interesting work with relevant content. Because of the large developments in the area, the paper would benefit from a more detailed description of the underlying physics with a proper comparison with the literature before publication.

Reviewer #4 (Remarks to the Author):

Authors of the paper entitled "Turnkey photonic flywheel in a Chimera cavity" present a chimera cavity within which they integrate a compact fiber Fabry-Perot. The fiber-based gain medium within the chimera cavity supplies energy for a Brillouin laser that then acts as a pump for a Kerr comb. This work builds on their prior demonstration (PRL 2020 seen as Ref. 8) which explores external pumping of a Brillouin laser-driven Kerr comb as a means of obtaining improved phase noise and better frequency stability. The primary mechanisms for improved performance are (1) the pump frequency noise suppression supplied by the internal Brillouin laser dynamics and (2) a reduction in photothermal noise produced competing effects within the resonators. The results are impressive and the relatively repeatable self-starting of the stable comb generation within this chimera cavity appears to be a substantive advance toward the turn-key operation of such novel laser systems. For these reasons, I believe that this work has the potential for high impact and will be of significant interest to the broad readership of Nat. Comm. For these reasons, I support the publication of this work in Nat. Comm. However, I do have a few points of clarification that I think the authors should address prior to publication below.

Comments:

- The following statement was unclear "Of note, our deterministic turnkey process is independent of 980-nm laser power of the EDFA in an offset range of ± 100 mW, where the reported single LCS attractor can be destroyed [30]." In particular, it was unclear to me what is meant by "an offset range of ± 100 mW" in this context. Please clarify this point.
- The authors identify a number of applications that this system could enable. Many of these applications rely heavily on the production of stable, low phase noise frequency combs. Through their measurements the authors quantify the phase noise at a 1MHz offset frequency. However, the close to carrier (< 1 MHz) phase noise is also very important for these applications. I think it would also be good to discuss the phase noise at lower offset frequencies and perhaps discuss the mechanisms that contribute to this reduced phase noise.
- To provide context, please make a brief comparison of the noise characteristics obtained with this system and compare with Kerr combs systems of similar size-scale.
- Also, a brief comparison to their earlier work in PRL seen as reference 8 could be instructive.
- To build on this work, I think it would be invaluable to include estimates and a discussion of noise sources and their contributions to the dynamics of this system. For this and future works I think it would be incredibly valuable to develop models that include photothermal effects brew on dynamics, nonlinearities, etc. Certainly to include all of these would substantially add to the length of the paper, and may not be reasonable. Nevertheless, progress in this direction I think would be invaluable to understanding the true potential of such hybrid laser systems.

Reviewer #2 (Remarks to the Author):

In their manuscript titled "Turnkey photonic flywheel in a Chimera cavity," the authors proposed and demonstrated the turnkey operation of Brillouin-Kerr soliton frequency combs using a Chimera cavity configuration, which consists of an FP microresonator nested in a fiber ring cavity. The concept is both fascinating and novel, as it breaks the traditional paradigm that the pump and the comb are co-generated in the same cavity in Brillouin-Kerr frequency combs. The decoupling scheme, along with the two-step pumping method, enables reliable turnkey operation, which is highly valuable for practical applications. Besides, the high coherence of the Brillouin-Kerr frequency combs demonstrated here can benefit areas such as ultrafast optics and microwave electronics.

Overall, the manuscript is well-written and detailed, and the experimental demonstrations are of exceptional quality. In my opinion, this work aligns well with the scope of Nature Communications, given its novelty and potential for broad interest. With a few minor modifications, I am convinced that this paper is suitable for publication in the journal.

Response: We thank Reviewer #2 for recognizing the importance of our robust turnkey soliton in a microresonator-filtered laser with improved noise performance at the photonic flywheel level.

The following are some places that may need further clarification:

1. I noticed that the authors refer to the Brillouin-DKS frequency comb with sub-optical-cycle timing jitters as a "photonic flywheel." However, I found that apart from the authors' previous work and ref. [7], there are few other papers that use this term. I would request the authors explain why the pristine temporal periodicity with sub-optical-cycle timing jitter is referred to as a photonic flywheel.

Response: The term "flywheel" is commonly used in mechanics, electronics, and optics [R1-R5]. A flywheel is a mechanical device which uses the conservation of angular momentum to store rotational energy. Even after the control stimulus has been removed, the flywheel can continue the rotation smoothly due to the stored mechanical energy. Therefore, it can be used as a reference in synchronous systems especially at short time scales.

The steady-state pulse propagating inside a mode-locked laser or a microresonator is the optical equivalent of a mechanical flywheel. Even without any stabilization, a mode-locked laser can have a very low phase noise level at high offset frequencies, making it very suitable as a flywheel oscillator and very precise timing reference. Here for our turnkey and self-stabilized soliton in the microresonator-filtered laser, the integrated jitter from 18 kHz to 1 MHz is such low (< 1 fs) that it can be viewed as a flywheel oscillator and precise timing reference. Of note, the phase noise at the low offset frequencies (< 20 kHz) can be easily reduced by an active feedback loop. Additionally, the fundamental comb linewidth is only 100 mHz so that each comb line can be viewed as a well-stabilized laser and flywheel oscillator. Therefore, our turnkey soliton microcomb with both ultralow comb linewidth and ultralow timing jitter can be referred to as a photonic flywheel.

[R1] <https://en.wikipedia.org/wiki/Flywheel>

[R2] Chambon, D., et al. "Design and realization of a flywheel oscillator for advanced time and frequency metrology." Review of Scientific Instruments 76(9) (2005): 094704.

[R3] Millo, J., et al. "Flywheel oscillator for atomic fountain clocks using ultra-stable lasers and a fiber-based optical frequency comb." 2009 IEEE International Frequency Control Symposium Joint with the 22nd European Frequency and Time forum. IEEE, 2009.

[R4] https://www.rp-photonics.com/optical_frequency_standards.html

[R5] https://www.rp-photonics.com/timing_jitter.html

2. The authors state that "From the quantum noise perspective, fiber cavity is a superior platform as its weaker mode confinement and lower Kerr nonlinearity preferably minimize the quantum limit towards sub-femtosecond DKS timing jitter in microsecond time scale." However, in a recent publication in Optics Letters, Vol. 46, Issue 7, pp. 1772-1775 (2021), quantum-limited DKS timing jitter was also achieved in a WGM cavity with a similar level of timing jitter to that of the present work. Could the authors comment on this?

Response: The quantum noise limited SSB phase noise power spectral density (PSD) of the soliton repetition rate is given by [23]

$$L_{\phi_{\text{QN}}}(f) = \frac{\sqrt{2}\pi}{2} \sqrt{\frac{\gamma}{\Delta_0(-D)}} \frac{g}{\eta\gamma^2} \times \left[\frac{1}{96} \frac{\gamma(-D)}{\Delta_0} \frac{\eta\gamma^2}{f^2} + \frac{1}{24} \left(1 + \frac{\pi^2 f^2}{\gamma^2} \right)^{-1} \frac{\eta\gamma^2}{\pi^2 f^2} \frac{\Delta_0(-D)}{\gamma} \right],$$

where 2γ is the resonance linewidth related to the Q factor, Δ_0 is the soliton detuning, g is the frequency shift of a resonant mode per photon, D is the normalized GVD of soliton mode, and η is the quantum efficiency of the detector. Since $g = \frac{\hbar\omega_0^2 cn_2}{Vn_0^2}$, where \hbar is the Plank constant, ω_0 is the soliton center frequency, c is the speed of light, n_2 is the Kerr nonlinear coefficient, V is the mode volume and n_0 is refractive index, **the most effective way to lower the DKS quantum noise limit is to decrease the Kerr nonlinear coefficient and increase the mode volume [27].**

The paper that the reviewer mentioned in the ref [25] cited in our manuscript. While our GIF-FP microresonator indeed provides a phase noise that is 8 dB better than that in ref [25], we realize that our statement that *fiber cavity is a superior platform* may not be rigorous as one can claim that the WGM cavity mode area can also be further increased to lower the quantum noise limit. We thus decided to remove the statement from the Introduction.

Platform	n_2 (m ² /W)	FSR (GHz)	Mode area	SSB phase noise (dBc/Hz, scaled to 10 GHz)	
				10 kHz	100 kHz
GIF FP microresonator [this work]	3.6×10^{-20}	~ 10	$255 \mu\text{m}^2$	-128	-147
silica wedge microresonator [25]	3.6×10^{-20}	~ 11	unknown	-120	-139

Revisions made: We have removed the statement "From the quantum noise perspective, fiber cavity is a superior platform as its weaker mode confinement and lower Kerr nonlinearity preferably minimize the quantum limit towards sub-femtosecond DKS timing jitter in microsecond time scale [8,9,27]." from the revised Introduction.

3. The paper provides a rather brief explanation of the turnkey principle. Could the authors provide the derivation of the attractor curve, the thermally stable regime (dashed box), and the boundaries of the various soliton states in Fig. 2a?

Response: Figure 2a is general schematic to show the turnkey principle in a simple way. It contains two parts of information: (i) the route of how the system is attracted to the specific point in the parameter space that can generate thermally stable SBL soliton; (ii) the microresonator temperature (MRT) is a key parameter that can change the final parameters especially the SBL detuning which can change the soliton state and deterministically select the soliton number.

Derivation of the attractor curve Of note, Fig. 2a is a representative schematic only, and does not account for many experimental details during the turnkey process as shown in Fig. 2b, 2c and 2d. For the stable SBL soliton, the final pump detuning is near zero while the final SBL detuning is at red side. However, the initial pump detuning at the beginning of the turnkey process can be either red-detuned or blue-detuned, depending on the system initial condition for example the large fiber cavity length. Figure 2a shows the case with initial red-detuned pump, which can be verified from the experimental PDH signal in Fig. 2d. The initial red-detuned pump corresponds to the initial red-detuned SBL due to the designed SBS frequency shift for SBL soliton generation. **In the revised Supplementary Information Section III, we have developed a comprehensive model and numerically reproduced the soliton attractor that matches well with Fig. 2a.**

In the figure below, we also show the turnkey soliton dynamics with initially blue-detuned pump. Therefore, our turnkey soliton is independent with the initial pump detuning or the large fiber cavity length, which is also verified by the experimental results in Fig. 3b.

Turnkey soliton dynamics with initial blue pump detuning. From top to bottom, real-time evolution of pump mode power, Brillouin mode power, comb power and pump detuning. Right panels are the zoomed-in traces in the time range from -1 s to 2 s.

Thermally stable regime For a given MRT, according to experimental results, the pump detuning fluctuation for thermally stable SBL soliton is ~ 200 kHz, while the pump frequency fluctuation is ~ 200 MHz. We estimate the SBL detuning regime to be ~ 40 kHz from pump detuning noise suppression (see revised Supplementary Information Section VII) for maintaining the soliton state.

Various soliton states In the revised Supplementary Information Section III, we show that the MRT range for stable SBL soliton generation is ~ 0.8 K. By changing MRT, we can achieve deterministic soliton number selection from 1 to 3. The temperature step is around 0.3 K, which depends on the temperature sensitivity of the SBS frequency shift.

Revisions made: The attractor curve, the thermally stable regime (dashed box), and the boundaries of the various soliton states in Fig. 2a strongly depend on many global parameters. Therefore, it is difficult to obtain the theoretical derivations. On the other hand, we include numerical analysis for some specific cases to help readers better understand the formation of soliton attractor in the revised Supplementary Information Section III.

4. The authors mentioned that the MRT is an effective control parameter that deterministically select the DKS soliton number. Could the authors provide a specific derivation for this?

Response: In short, the soliton existing range is non-degenerate due to the compensated thermal nonlinearity by the two coexisting lasers in the microresonator. Therefore, different SBL detunings results in different soliton states. The MRT can change the SBS frequency shift, and consequently the SBL detuning and the soliton number. We have developed a comprehensive model and reproduced turnkey soliton in the revised Supplementary Information Section III. There, we show that the MRT range for stable SBL soliton generation is ~ 0.8 K. By changing MRT, we can achieve deterministic soliton number selection from 1 to 3. The temperature step is around 0.3 K, which depends on the temperature sensitivity of the SBS frequency shift.

Revisions made: We have also added the explanation for the deterministic selection of soliton number in Section II of the Supplementary Information as

“Figure S4 shows the soliton number dynamics when changing the microresonator temperature in the soliton existing regime. The ECDL is swept from blue side to red side. By changing the microresonator temperature, the SBS frequency shift is changed thus both the SBL detuning and thermal equilibrium condition are changed when pump detuning is near zero. With two intracavity lasers compensated thermal effect, the SBL soliton existing regime is no longer degenerate due to the XPM effect [4–7]. The soliton steps showing the soliton number switching are clearly observed in Fig. S4. By changing the temperature, we can deterministically select the soliton number from 3 to 1 before pump power drops or pump laser exits the microresonator at what time the pump detuning is around zero detuning. Therefore, in our microresonator-filtered fiber laser, we can deterministically select the soliton number by changing the microresonator temperature.”

Fig. S4. Soliton dynamics by sweeping the pump laser frequency under different microresonator temperatures. (a) the final soliton number (before pump power drops or pump laser exits the microresonator) is 3; (b) the final soliton number is 2; (c) the final soliton number is 1.”

5. How the repetition rate frequency evolution in Fig. 3a was measured?

Response: In the Methods under the Measurement of comb repetition rate phase noise and timing jitter, how the repetition rate frequency evolution in Fig. 3a was measured is described. “The RF beat note of the soliton repetition rate is directly detected by a fast photodetector (EOT, ET-3500F) and measured by an electrical spectrum analyzer (E4407B, Keysight) at 5 Hz to show its evolution.”

6. It seems that the schematic in Fig. 1a is missing the 980 nm pump laser.

Response: To draw reader’s attention to the key components for the turnkey soliton, we intentionally neglected the EDFA details in the schematic to avoid distraction. Indeed, 980-nm pump lasers are required for the EDFA.

Revisions made: We have revised the description in the Methods as “The home-made EDFA is bidirectional pumped, consisting of a 2-m highly Erbium-doped fiber (SM-ESF-7/125, Nufern), two 980/1550 nm wavelength demultiplexer (WDM) and two 980-nm diode lasers for each direction.”

7. In Fig. 5a, the linewidths of all the lasers are significantly lower than the corresponding linewidths in the WGM microresonator [Phys. Rev. Lett. 126, 063901 (2021)]. Could the authors provide a quantitative analysis of this difference?

Response: The SBL and its comb linewidth narrowing factor r is determined by $r = (1 + \Gamma_b / \gamma)^2$, where Γ_b is the Brillouin gain bandwidth and γ is the Brillouin mode cavity linewidth. Therefore, a larger Q factor can lead to a better SBL noise suppression and a larger linewidth narrowing factor.

First, we compare the two similar externally pumped cases: our previous work [Nature Communications 13 (1), 6395] and the case in Phys. Rev. Lett. 126, 063901 (2021) (PRL), as listed in the table below. We attribute the linewidth difference mainly to the difference of the Q factor or the Brillouin mode cavity linewidth, as indicated by the above-mentioned conclusion. Of note, the microresonator in our previous NC paper is almost the same with the one in the current turnkey soliton work.

Paper source	Q factor (Million)	Estimated SBS gain bandwidth Γ_b (MHz)	Cavity linewidth γ (MHz)	Calculated Linewidth narrowing factor r	Experimental Linewidth narrowing factor r	Experimental Fundamental linewidth (Hz)
NC	~200	20	~1	440	325	~0.4
PRL	44.4	25	4.4	44	42	23.5

Second, for our turnkey soliton case, the pump laser phase noise suppression in our microresonator-filtered laser can be viewed as a result of self-injection locking. The ultrahigh Q of our MMF FP microresonator enables the ultralow fundamental linewidth of 0.1 Hz for the pump laser. We also find that the SBL fundamental linewidth is almost the same with the pump laser since both are limited by the pump RIN.

Revisions made: We have added the comprehensive noise analysis and comparison in terms of the laser comb linewidth, in the revised Supplementary Information Section VII to Section X.

8. In Fig. 5c, the SBS frequency shift follows a $1/f^3$ trend, while the SBS comb repetition rate follows a $1/f^2$ trend, and the noise level of the SBS comb repetition rate is much lower than that of the SBS frequency shift. Could the authors provide further analysis of these results?

Response: In short, the phase noise PSD with $1/f^3$ trend for the SBS frequency shift is mainly dominated by frequency flicker noise resulting from pump laser frequency drift. And the phase noise of the SBL soliton comb repetition rate following $1/f^2$ trend, is dominated by the SBL RIN or pump RIN. Since the pump and SBL are not fully phase locked but only share 10-dB common mode noise suppression, the coherence between pump and SBL is worse compared to the coherence between phase-locked SBL comb lines. Therefore, the noise level of the SBS comb repetition rate is much lower than that of the SBS frequency shift.

Revisions made: We have added the comprehensive noise analysis in the revised Supplementary Information Section VII to Section X.

Reviewer #3 (Remarks to the Author)

The paper ‘Turnkey photonic flywheel in a Chimera cavity’ by Nie and co-workers deals with the self-starting of dissipative Kerr solitons (DKS) in a microcavity filtered fibre laser.

The paper leverages the Brillouin effect and I find that it provides an important capability for the field. The authors, together with other groups (e.g. [23]), are establishing important results in the use of the Brillouin effect in microresonators. In this regard, the recent work of Baicheng Yao and coworkers, Phys. Rev. Lett. 130, 153802, should probably be cited too.

Response: We thank Reviewer #3 for recognizing the importance of SBS effect for dissipative Kerr soliton generation. The recent work [Phys. Rev. Lett. 130, 153802] by Baicheng Yao and coworkers demonstrating multiple combs generation from cascaded Brillouin effect was not published and not available during our peer review process.

Revisions made: We have added the reference in the Introduction as Ref. [26].

I find also very interesting the use of the Brillouin effect in a microcavity-filtered configuration. The authors explored a microcavity-filtered configuration already in their recent work (Light: Science & Applications volume 11, Article number: 296 (2022)). Differently from this, here the authors use an intracavity narrowband filter to isolate only a continuous wave that lases in the whole system sustaining the soliton in the Kerr cavity.

Response: We thank Reviewer #3 for the interest in ultralow-noise soliton generation with Brillouin effect in the microresonator in a microresonator-filtered laser. We will show in the later responses that the physical mechanism of this work is distinctly different from our previous work [Light: Science & Applications 11, 296 (2022)] and a more recent work by another group [Nature 608, 303 (2022)] even though there are similarities in the optical setups.

I have to say, however, that the authors appear to ignore a good chunk of literature on microresonator-filtered lasers. These approaches have been largely investigated for more than 10 years, see Nature Communications 3, 765 (2012), Nature Photonics 13 (6), 384-389 (2018). The configuration using a narrow filter to allow only a CW pump to oscillate was indeed proposed in 2013, Optics Express 21 (11), 13333-13341, and largely used for quantum microcomb generation, e.g. Nature Communications 6 (1), 8236.

Response: In our original manuscript, we neglected these references of microresonator-filtered lasers because we wanted to focus on the literature that is directly relevant to the turnkey low-noise soliton generation. This is why we also did not cite our own previous microresonator-filtered laser work [Light: Science & Applications 11, 296 (2022)].

Revisions made: We have added these references in the Introduction as Refs. [27–36].

The authors call the microcavity-filtered approach a ‘Chimera cavity’. I find that this confuses the literature. ‘Chimera’ is a well-known term in the nonlinear laser literature to describe a specific type of state (chaotic + stable state, see, e.g. ‘Laser chimeras as a paradigm for multistable patterns in complex systems’ Nature Communications volume 6, Article number: 7752 (2015)), completely out of context here. I would recommend keeping using the established terminology and changing the title to ‘Turnkey solitons in a microresonator-filtered laser via Brillouin effect’.

Response: Thanks for the reviewer’s suggestion. “Chimera cavity” is indeed a new terminology and there were internal debates within the group before the manuscript submission. As the reviewer mentioned, “Chimera state” is a well-known term in nonlinear dynamics that describes states of coexistence of synchronous and asynchronous motion or the coexistence of coherent and incoherent dynamics. Our understanding is that “Chimera” is an adjective that conveys the idea that a system is composed of distinctly different components. Our microresonator-filtered laser consists of a comb-generating passive Fabry-Perot microcavity nested in a pump-generating active ring cavity. These two nested cavities not only differ in their topology but also in their function. Although we believe “Chimera cavity” highlights this fact, we decide to take the reviewer’s suggestion and change it to “microresonator-filtered laser cavity” throughout the paper to avoid any potential confusion.

Revisions made: We have revised the “Chimera cavity” as “microresonator-filtered laser” throughout the manuscript and the title has been changed to “Turnkey photonic flywheel in a microresonator-filtered laser”.

In their introduction, the authors compare their approach only to self-injection locking (SIL). They state: ‘Unlike nonlinear SIL, our approach does not suffer from the vulnerability to feedback phase fluctuation and enables the deterministic selection of DKS soliton numbers. In addition, we demonstrate the self-healing capability of returning to the original comb state from instantaneous perturbations. More importantly, our approach allows the access to the ultralow noise comb state’. These properties have all been demonstrated already with microresonator-filtered microcombs, in laser-cavity soliton configuration. This should be properly acknowledged.

Conversely, well after their introduction, they note: ‘Of note, the Chimera cavity shares some similarity with the self-emergent laser cavity-soliton (LCS) [30] but the working principles are distinctly different. Self-emergent LCS critically depends on the parameters of the outer laser cavity which is more susceptible to environmental disturbance. Ultralow comb noise and long-term stability have not been demonstrated with the self-emergent LCS approach.’ While I am not convinced that the principles are distinctly different, it is surely not correct that ‘Ultralow comb noise and long-term stability have not been demonstrated’. Beyond Ref [30], where the system was shown to operate on the hour timescale, the metrological properties have been more recently discussed in Applied Physics Letters 122 (12), 121104, showing both low noise generation, long term, state of the art free running stability.

It is incorrect that ‘Self-emergent LCS critically depends on the parameters of the outer laser cavity which is more susceptible to environmental disturbance,’ Applied Physics Letters 122 (12),

121104 shows that the external parameters can be largely modulated while retaining the solitary generation and stability properties.

The authors should, hence, avoid justifying the novelty of their work with unfactual statements on demonstrated results and instead provide an interpretation of their data that highlights the critical differences and possible opportunities.

Response: Even though we do not agree with the reviewer that we are making unfactual statements (and we will provide our reasonings), we take the reviewer's suggestion to focus on our results and include summaries of different microcomb noise performances in the revised Supplementary Information Section X. After all, our system is based on the principle of Brillouin-DKS which is fundamentally different from the self-emergent LCS where Brillouin lasing is not involved.

Revisions made: In the revised Introduction, we have acknowledged that LCS in a microresonator-filtered laser configuration has also achieved the phase-insensitive turnkey operation, the soliton self-healing, and the deterministic selection of soliton numbers. We have removed any unnecessary direct comparison with LCS in the main text.

Comb repetition rate phase noise

In the APL paper, the SSB phase noise of the comb repetition rate is -101.3 dBc/Hz at 100 kHz and -115.8 dBc/Hz at 1 MHz (both scaled to 10 GHz carrier frequency). In our manuscript, the SSB phase noise of the comb repetition rate are -147 dBc/Hz at 100 kHz and -166 dBc/Hz at 1 MHz (both scaled to 10 GHz carrier frequency). Therefore, our turnkey Brillouin-DKS has 40 dB better phase noise than the demonstrated LCS.

Comb fundamental linewidth

We have shown that the fundamental linewidth of our turnkey Brillouin-DKS is only 100 mHz thanks to the SBS linewidth narrowing effect. The comb fundamental linewidth of LCS has not been measured yet. In the revised Supplementary Information Sections VII-IX, we have included detailed noise analyses summarizing the benefits of the noise-suppressing SBS process in the turnkey Brillouin-DKS.

Long-term stability

In Ref. [30] and the APL paper, the authors only show the long-term stability of ~20 minutes, not "on the hour time scale". However, we show the long-term stability of 2 hours for our turnkey Brillouin-DKS in Fig. 5d. The long-term stability in both cases is mainly determined by the stabilities of pump frequency and pump detuning. In our case, the SBS process can enhance the stability of SBL frequency and SBL detuning from the pump laser which circulates in the large fiber cavity. Therefore, our turnkey DKS generated by SBL shows better long-term stability than the turnkey LCS even when most of the setup is not enclosed for shielding the environmental noise. And this is the reason why we shifted from our previous self-starting LCS work in [Light: Science & Applications 11 (1), 296], which is the same as the APL paper and Nature paper, to the present turnkey Brillouin-DKS.

Modulation depth tolerance

The table below compares the tolerances of LCS and turnkey Brillouin-DKS to different types of modulations. The turnkey Brillouin-DKS is more tolerant to all types of modulation thanks to the extended soliton existing range from the compensated thermal nonlinearity.

	APL paper	This work
Fiber cavity length change	$\pm 0.15 \mu\text{m}$	$\pm 4 \mu\text{m}$
Pump frequency change	$\pm 3 \text{ MHz}$	$\pm 80 \text{ MHz}$
Repetition rate change	$\pm 1 \text{ kHz}$	$\pm 5 \text{ kHz}$
980-nm pump power change	$\pm 9 \text{ mW}$	$\pm 100 \text{ mW}$

Regarding the distinction between the working mechanism between this paper and the LCS, I think that there are many similarities and specific differences, which make the paper relevant. This requires, however, a fair comparison with what has been already demonstrated.

To me, this paper is an alternative pathway to control the slow mechanism that can induce soliton microcombs self- emergence in two cavities system, as demonstrated in [30]. Although the Brillouin effect has been explored already with externally pumped sources, with the mechanism that the authors summarise in their extended Fig. 1 and previously, e.g. in [23], its use in a microresonator filtered configuration is not obvious and merit publication. Before being ready for it, there are however some points that need to be clarified, together with a proper amendment and a fair discussion of the literature.

In terms of slow-fast nonlinear interaction, the authors use a similar principle of self-emergent LCS. They force a soliton state to be the dominant attractor of the system and, to this aim, they act on a global parameter to balance the nonlinearities of the system. This is the physical concept developed in [30], where the global parameters were cavity loss and pump energy. Here, specifically, they use the temperature: ‘When the microresonator temperature (MRT) is set correctly such that the offset frequency is slightly larger than the SBS frequency shift, the co-existing blue-detuned pump and red-detuned SBL in the microresonator manifest itself into the thermally stable DKS attractor.’ This is the same principle used in [30], but with an interesting twist, because the Brillouin effect was not present in [30]. The Brillouin shift works to compensate for the thermal nonlinearity of the microcavity and brings the system to work on the red-detuned slope of the Kerr resonator, inducing the solitons. I note, however, that this was already discussed in [23] and with the two-step pumping in [8,9]. The authors report improved robustness by using the microcavity nested within the amplifying cavity. Hence, I would suppose that the amplifying cavity plays a role here.

Interestingly, the authors state that they observe a balance between the nonlinear shift induced by the Brillouin effect in the microcavity and the thermal effects in the erbium fibre. The authors comment ‘While the DKS attractor state is mainly defined by the microresonator, the turnkey dynamics closely follows the optical pathlength change of the active ring cavity that is caused by refractive index increase and thermal expansion resulting from the EDFA pump absorption [32–34].’ The authors follow the same citations [32–34] discussed in [30] to highlight this phenomenon. In case of the LCS, it has been demonstrated that this effect balances the nonlinearity in the

microcavity to induce the soliton. The authors should then quantify ‘the optical pathlength change of the active ring cavity’ against the Brillouin shift and thermal microcavity nonlinearity to provide a proper physical insight in the principle, otherwise, there is no evolution from previous work on Brillouin states, e.g. [23], nor a distinct novelty compared to [30].

Response: A successful turnkey operation requires both soliton self-excitation and soliton self-stabilization mechanisms. For self-emergent laser cavity soliton [30], soliton is self-excited by laser self-organization process in the microresonator-filtered laser and self-stabilized by the balance between the gain (or loss) nonlinearity in the active fiber cavity and thermal nonlinearity in the microresonator. For our turnkey Brillouin-DKS, while soliton is also self-excited by laser self-organization process in the microresonator-filtered laser, it is self-stabilized because of the thermal nonlinearity compensation by the co-existing blue-detuned pump and red-detuned SBL in the microresonator. We have developed a comprehensive model and reproduced turnkey soliton in the revised Supplementary Information Section III. The numerical modeling confirms that the self-stabilization of our turnkey Brillouin-DKS does not depend on the slow-fast nonlinear interaction and the SBS frequency shift, instead of the laser gain, is the dominant factor that defines the soliton attractor.

We did not say that the thermal balance is achieved between the nonlinear shift induced by the Brillouin effect in the microresonator and the thermal effects in the Er-doped fiber. What we said is that the turnkey dynamics (or self-excitation dynamics) closely follows the laser turn-on process in the active ring cavity that serves as a spontaneous scanning of pump detuning to drive the system evolution into the DKS attractor state. During the laser turn-on process, optical pathlength change of the active ring cavity, caused by refractive index increase and thermal expansion resulting from the EDFA pump absorption, is the mechanism that leads to the spontaneous scanning of pump detuning. The numerical modeling in the revised Supplementary Information Section III also confirms that the laser gain plays a key role in soliton self-excitation but not in the soliton self-stabilization. On the other hand, co-existence of blue-detuned pump and red-detuned SBL is responsible for thermal nonlinearity compensation and is the self-stabilization mechanism.

Based on these analyses, we thus do not think the mechanism of our turnkey Brillouin-DKS can be explained using the framework of Ref. [30].

Revisions made: We have developed a comprehensive model and reproduced turnkey soliton in the revised Supplementary Information Section III.

III. Theoretical and numerical analysis

First, we model the formation and dynamics of the Brillouin-Kerr frequency comb for the externally pumped case to verify the importance of the Brillouin frequency shift in the two-step pumping scheme. We employ a set of coupled mode equations in frequency domain which combines both the three-wave interaction for the SBS process and four-wave-mixing in both pump and Brillouin mode families during the comb-generating process. We also consider the thermal effect in our FFP microresonator, which causes the thermo-optic drift of the mode resonance. Therefore, the theoretical model of Brillouin-Kerr resonator takes the form:

$$\frac{\delta \tilde{P}_\mu}{\delta t} = \left(-\frac{\gamma_p}{2} + i\sigma_p - i\beta_p \Delta T + i[D_{p,\mu}] \right) \tilde{P}_\mu - ig_b \delta_0 b B_0 - ig_{p\kappa_1} \mathcal{F}^{-1}\{|P|^2 P\}_\mu - i2g_{\kappa_2} \mathcal{F}^{-1}\{|B|^2 P\}_\mu + \delta_0 \sqrt{\kappa_p} S_{in}, \quad (S1)$$

$$\frac{\delta \tilde{B}_\mu}{\delta t} = \left(-\frac{\gamma_B}{2} + i\sigma_B - i\beta_B \Delta T + i[D_{B,\mu}] \right) \tilde{B}_\mu - ig_b \delta_0 P_0 b^* - ig_{B\kappa_1} \mathcal{F}^{-1}\{|B|^2 B\}_\mu - i2g_{\kappa_2} \mathcal{F}^{-1}\{|P|^2 B\}_\mu, \quad (S2)$$

$$\frac{\delta b}{\delta t} = \left(-\frac{\Gamma_b}{2} + i\sigma_b + i\Delta\Omega \right) b - ig_b \delta_0 P_0 B_0^*, \quad (S3)$$

$$\frac{\Delta T}{\delta t} = -\frac{1}{\tau_T} \Delta T + c_{T,p} \sum_\mu |\tilde{P}_\mu|^2 + c_{T,B} \sum_\mu |\tilde{B}_\mu|^2, \quad (S4)$$

where \tilde{P}_μ , \tilde{B}_μ , b are the fields of pump mode, Brillouin mode and acoustic wave, respectively (μ is the mode number). $\gamma_{p(B)}$ is the photon decay rate for pump (Brillouin) modes including both intrinsic and external losses. Γ_b is the phonon decay rate. g_b is the coupling coefficient between the optical and acoustic modes in the SBS process, while $g_{p(B)\kappa_1}$ and g_{κ_2} are the self and cross phase modulation coefficients respectively. The dispersion $D_{p(B),\mu}$ relates the photon mode frequency, $\omega_{\mu,p(B)}$, to the center frequency, $\omega_{0,p(B)}$, by the relation $\omega_{\mu,p(B)} = \sigma_{p(B)} + \omega_{0,p(B)} + D_{p(B),\mu}$. The coupled equations are with reference to the detuning values $\sigma_{p(B)} = \omega_{p,B} - \omega_{0,p(B)}$ and $\sigma_b = \Omega_b - \Omega_0$ subject to the phase matching relationship $\sigma_p - (\sigma_B + \sigma_b) = 0$. The term $\Delta\Omega$ represents the frequency offset between the SBS wave and the central Brillouin mode frequency. To facilitate numerical simulation the inverse Fourier, transform operator, \mathcal{F}^{-1} , is used to calculate the four wave mixing processes between pump and Brillouin modes. κ_p and S_{in} correspond to the coupling rate and power of the external pump. The fourth equation describes the intracavity power induced temperature change ΔT from the power absorption. In the equation, $c_{T,p(B)}$ and τ_T are the modal thermal absorption coefficient and thermal relaxation time respectively. The temperature change will cause the frequency drift of the pump and Brillouin mode due to the refractive index change through thermo-optic effect. The frequency shift can be expressed as $\Delta\omega_{T,p(B)} = \beta_{p(B)} \Delta T$, where $\beta_{p(B)}$ is the thermo-optic coefficient.

Using the above set of equations, we numerically calculate 2048 modes of both the Brillouin and pump mode families for the *externally* driven thermally stable Brillouin Kerr-Soliton generation, as shown in Fig. S5. The parameters used for simulation are: $\kappa_p = \kappa_B = 2\pi * 0.4$ MHz, $\gamma_{0,p} = \gamma_{0,B} = 2\pi * 0.7$ MHz, $\Gamma = 2\pi * 30$ MHz. The mode frequencies were defined $\Omega = 2\pi * 10.5$ GHz and $\omega_p = 2\pi * 193.73$ THz. The dispersion parameters were set for the FSR as $D_{1,p} = D_{1,B} = 10$ GHz, GVD $D_{2,p} = D_{2,B} = 0.57$ kHz and high-order dispersions are neglected. $c_{T,p} = c_{T,B} = 50$ K/J · GHz, $\gamma_{abs,p} = \gamma_{abs,B} = 5$ K/J, $\beta_p = \beta_B = 2\pi * 100$ kHz/K, and $1/\tau_T = 10$ MHz. $V_{eff,B} = V_{eff,p} = 3.181E^{-12}$ μm^3 .

Figure S5a shows the evolution of stable Brillouin-DKS generation process in the presence of thermo-optic drift when pump frequency is scanned from the blue side to red side. If the SBS frequency shift is set to be slightly larger than the offset frequency ($\Delta\Omega$) between pump and Brillouin center mode resonances, a blue-detuned pump and red-detuned SBL can stably coexist in the microresonator. When the soliton is generated, the intracavity power won't drop as in Fig. S1d but will increase (Fig. S5a) due to the two coexisting lasers, leading to thermal nonlinearity compensation. The numerical results in Fig. S5a agree with the experimental ones in Fig. S1b, verifying the accuracy of our model. If the SBS frequency shift is set to be smaller than the offset frequency ($\Delta\Omega$) between pump and Brillouin center mode resonances, the SBL is blue-detuned, which leads to chaotic comb generation or continuous-wave generation, as shown in Fig. S5b and S5c. Of note, in the experiment the relationship between the SBS frequency shift and the offset frequency ($\Delta\Omega$) is adjusted by the microresonator temperature which mainly changes the SBS frequency shift. However, in the simulation, the relationship is adjusted only by changing the offset frequency ($\Delta\Omega$) in Eq. (S3).

Fig. S5. Dynamics of externally pumped Brillouin-Kerr microresonator when pump is scanned from blue side to red side. The pump detuning is normalized by the cavity linewidth of pump mode. (a) Soliton generation; (b) chaotic comb generation; (c) continuous-wave generation. The first row shows the evolution of intracavity energy (for both center modes, red: SBL, blue: pump) and the intracavity temperature (magenta). The second row shows the evolution of SBL detuning (normalized by the cavity linewidth of Brillouin mode). The third and fourth rows show the spectral and temporal evolution of Brillouin mode families, respectively.

Next, we model the microresonator-filtered laser to numerically verify the formation of soliton attractor. To form a laser cavity, the input and output of the microresonator are connected through the active gain and a narrowband spectral filter. Since only the pump center mode (single frequency) \tilde{P}_0 experiences laser gain, the saturable gain g reads as

$$g = \frac{g_0}{1 + \frac{\kappa_p |\tilde{P}_0|^2}{P_{sat}}} \quad (\text{S5})$$

where g_0 represents the small signal gain of the fiber amplifier and P_{sat} is the saturation power of the EDFA. Additionally, the refractive index change of the EDFA, governed by the excited pump ions, can cause the lasing frequency shift in the large fiber cavity, which can equivalently act as a frequency-sweeping laser as in the above-mentioned sweeping case. Therefore, the shifted pump frequency will lead to the pump detuning change, which is modelled as $\sigma_p = |s_{in}|^2 \xi + \sigma_{p,0}$, where ξ is a fitting parameter set to model the external cavity red shift and $\sigma_{p,0}$ is the initial cold cavity detuning of the pump mode. The microresonator-filtered laser cavity model is completed by replacing the external pump term in Eq. S(1) with the amplified power of pump center mode $|s_{in}| = \sqrt{g|P_0|^2}$. In the simulation, we set the small signal gain to $g_0 = 2.55$ 1/m and $P_{sat} = 1$ W for a 2-m EDFA.

Numerical demonstration of soliton attractor As shown in Fig. S6a-c with $\beta_p = \beta_B = 2\pi * 100$ kHz/K, Brillouin-DKS can self-start from seeded quantum noise. The pump is spontaneously attracted to near the resonance peak for maximizing the output power. After some oscillations (power

oscillations in Fig. S6a, horizontal oscillations in Fig. S6c and animation Video S1 online) caused by both the gain effect and thermal effect, the Brillouin-DKS is eventually attracted to the stable magenta star point in the detuning diagram (Fig. S6c), where pump is blue detuned and SBL is red detuned. The numerical result in Fig. S6c matches well the schematic soliton attractor in the main text. Additionally, we numerically simulate the case with a 10 times larger thermo-optic coefficient. As shown in Fig. S6c-e, the larger thermal nonlinearity can be well compensated, and the system can eventually evolve into the stable Brillouin-DKS state (green star point in Fig. S6c). This verifies that our microresonator-filtered laser configuration is general and can be applied to different microresonators.

Fig. S6. Numerical demonstration of self-starting soliton in a microresonator-filtered fiber laser. (a)(e) Energy evolution of both pump (blue) and SBL (red), (b)(f) stable soliton spectrum, (c)(g) stable temporal waveform, and (d) the detuning evolution during the self-starting process to a stable soliton state. (a-c) $\beta_p = \beta_B = 2\pi * 100 \text{ kHz/K}$; (e-g) $\beta_p = \beta_B = 2\pi * 1 \text{ MHz/K}$.

Soliton attractor determined by the SBS frequency shift Our simulation model reveals that the cavity attractor is mainly determined by the SBS frequency shift, which finally changes the detunings and intracavity powers. Similar to Fig. S5, by adjusting the SBS frequency shift, the final stable cavity states vary from sing-frequency SBL generation to Brillouin-DKS generation as shown in Fig. S7.

Fig. S7. Cavity attractor under different SBS frequency shift (animation Video 2 online). (a)(e) Energy evolution of both pump (blue) and SBL (red), (b)(f) stable soliton spectrum, (c)(g) stable temporal waveform, and (d) the detuning evolution during the self-starting process to a stable soliton state. (a-c) $\Delta\Omega = -2\pi * 30 \text{ MHz}$; (e-g) $\Delta\Omega = -2\pi * 20 \text{ MHz}$.

Soliton attractor insensitive to the large fiber cavity change Different from the reported laser cavity soliton where the balance of the thermal nonlinearity in the microresonator is realized by the gain nonlinearity, for our Brillouin-DKS in the microresonator-filtered laser, the active gain does not play a role in the thermal nonlinearity compensation. In fact, the thermal nonlinearity is compensated only in the microresonator by the two coexisting pump and Brillouin laser. The large fiber cavity here only provides the equivalent single frequency pump sweeping process resulting from the refractive index increase and thermal expansion. Therefore, it allows self-starting cavity dynamics to be independent of the initial feedback phase or fiber cavity length.

By implementing the two-step pumping scheme in the microresonator, the generated Brillouin-DKS is insensitive to the large fiber cavity change, either the length change or gain change. The cavity length change of the large fiber cavity can lead to the pump detuning change. Thanks to the SBS process, the SBL detuning change can be largely reduced (see Section VII for details) from the pump detuning. Therefore, even strong pump frequency modulation (or strong cavity length modulation) does not adjust the final soliton attractor significantly.

The EDFA gain change will also lead to the pump detuning change as well as the intra-microresonator power change. Thanks to the thermal nonlinearity compensated by the two coexisting lasers, the system can tolerate a large pump power change through thermal self-organization. Therefore, the Brillouin-DKS can exist over a broad range of EDFA gain values. Experimentally, the EDFA pump power tolerance of our deterministic turnkey process is $\pm 100 \text{ mW}$ at 1.5 W . In simulation, we see that the integrated gain can change by $\pm 0.25 \text{ dB}$ while remaining in the same soliton state, which corresponds to a $\pm 6\%$ change in the EDFA pump power agreeing well with the experimentally observed $\pm 6.5\%$. In all, the Brillouin-DKS has strong perturbative immunity and excellent long-term stability.

Indeed, in [30] the balance of the thermal nonlinearity of the Kerr microcavity was completely obtained with the nonlinearity of the laser cavity only. It remains the question then, of how much

the external cavity plays a role in providing a more robust locking of the state also in the present work. Otherwise, it is unclear why the Brillouin cavity needs to be inserted in the amplifying loop and cannot simply be externally connected to the pump laser as in the Extended Fig. 1 and previous works.

Response: Thanks for the reviewer's suggestion. It is indeed very intriguing to consider the adoption of nonlinear self-injection locking to pump the Brillouin-DKS generation and compare its turnkey characteristics with the microresonator-filtered laser demonstrated in this work. We will design a follow-up study based on the reviewer's insightful suggestion.

The MRT temperature is the external parameter that allows the system to achieve the red-locking microcomb states, which is the key for the soliton generation. In the supplementary, they state that soliton generation occurs in a range of 0.8 K. This is a narrow range of temperature, which I would expect to be susceptible not only from environmental changes (what temperature stabilisation accuracy are they using on the microcavity?), but also from power changes within the system. Can the authors report the effective working temperature of the microcavity? Is it always the same? Can the author comment on the dependence of this range with respect to the circulating power within the laser? Do the authors observe a dependency on the losses of the system or on the power balance between the two lasing polarisation components?

Response: As shown in Fig. S2 in the revised Supplementary Information, the microresonator is well enclosed in a temperature-controlled holder with a 10-mK temperature stability. The MRT is not susceptible to either environmental or power changes. We also do not observe any obvious dependence of the soliton generation MRT range on either gain, loss, or pump power.

The authors comment 'Of note, our deterministic turnkey process is independent of 980-nm laser power of the EDFA in an offset range of ± 100 mW'. This number makes little sense if it is not related to the actual gain and average power circulating in the laser. Can the author please comment on the change of laser power that they obtain within the ± 100 mW change of their pump power? How much is the change of temperature in the microcavity and its resonance shift?

Response: The EDFA pump power tolerance of our deterministic turnkey process is ± 100 mW at 1.5 W, corresponding to $\pm 6.5\%$ of changes. The small-signal gain of EDFA is ~ 35 dB at 1.5 W. The microresonator resonance shift is measured to be ± 80 MHz within this EDFA pump power change. In the revised Supplementary Information Section III, we have developed a comprehensive model and reproduced such EDFA pump power tolerance. "By implementing the two-step pumping scheme in the microresonator, the generated Brillouin-DKS is insensitive to the large fiber cavity change, either the length change or gain change. The cavity length change of the large fiber cavity can lead to the pump detuning change. Thanks to the SBS process, the SBL detuning change can be largely reduced (see Section VII for details) from the pump detuning. Therefore, even strong pump frequency modulation (or strong cavity length modulation) does not adjust the final soliton attractor significantly."

The EDFA gain change will also lead to the pump detuning change as well as the intra-microresonator power change. Thanks to the thermal nonlinearity compensated by the two

coexisting lasers, the system can tolerate a large pump power change through thermal self-organization. Therefore, the Brillouin-DKS can exist over a broad range of EDFA gain values. Experimentally, the EDFA pump power tolerance of our deterministic turnkey process is ± 100 mW at 1.5 W. In simulation, we see that the integrated gain can change by ± 0.25 dB while remaining in the same soliton state, which corresponds to a $\pm 6\%$ change in the EDFA pump power agreeing well with the experimentally observed $\pm 6.5\%$. In all, the Brillouin-DKS has strong perturbative immunity and excellent long-term stability.”

On a minor point, I find the description of the Pound–Drever–Hall (PDH) modulation not clear. An additional figure to show these elements within the cavity would be helpful. The authors refer to a PDH signal and describe a MHz modulation of the intracavity pump. Usually, a PDH approach includes also an electronic feedback mechanism to lock the laser, e.g. on a filter slope or another laser. Here, however, such locking is not mentioned and seems not present. It appears that the authors only modulate the signal and use the MHz beating to track its changes against an ECDL reference laser. If it is so, calling this approach PDH is confusing. Could the authors comment on this and discuss if any other feedback mechanism against the laser power is used? The authors state ‘Thermal effect and Kerr effect are considered in the PDH signal measurement’. Can they please clarify how they take them into account in the measurement? Can the author comment on the stability of the ECDL laser used as the reference for this measurement?

Response: Thanks for Review #3’s suggestion. We have added the figure illustrating how we get the PDH error signal in the Supplementary Information Section XIII. Here we do not apply any feedback to the system, but only use the PDH error signal to study the dynamics. This method has been proved to be an effective characterization tool for DKS microcombs as shown in [*Nature Photonics* 8.2 (2014): 145-152; *Optica* 2.12 (2015): 1078-1085]. Since the PDH error signal measures the overall pump detuning, any resonance frequency shift induced by thermal and nonlinear effects will be captured in the PDH error signal. We have revised the Method section to make it clear. The ECDL used for measuring the pump laser frequency shift has a frequency stability of 2 MHz during the measurement time.

Revisions made: We have revised the Method section accordingly. “A phase modulator is inserted in the active fiber ring cavity for all the experiments to monitor the pump detuning via the open-loop PDH error signal (see more details in Supplementary Information Section XII).”

“Since the PDH error signal measures the overall pump detuning, any resonance frequency shift induced by thermal and nonlinear effects will be captured in the PDH error signal.”

“The pump laser frequency shift is monitored by the beat note between the pump laser and a tunable ECDL that has a frequency stability of 2 MHz during the measurement.”

In Supplementary Information XII, we have added the figure illustrating how we get the PDH error signal as “

XII PDH error signal

We show in Fig. S14 how we experimentally obtained the PDH error signal in the microresonator-filtered laser (a) and in the externally pumped configuration (b).

Fig. S14. Measurement of PDH error signal. (a) in the microresonator-filtered laser. (b) in the externally pumped configuration. PD: photodetector; PM: phase modulator; PS: phase shifter; LPF: low-pass filter.”

Regarding the capability of deterministically selecting the DKS soliton number, I believe that the authors do not explain clearly why they obtain them. It should be investigated further. This particular aspect is different from the LCS. In [30], the deterministic generation of a certain soliton number occurs because the lasing light is also pulsed, and the self-locking mechanism is energy dependent. This allows the discrimination of the soliton number. Here however the lasing pump is CW, like in a Lugiato-Lefever or a self-injection locking system. In these approaches, the soliton number, and also the relative position of the solitons within the crystal, are usually dependent on additional phase parameters, like in ref [35]. Considering that the authors obtain a two-soliton state with always the same phase, I would agree with the authors that this appears to be similar to [35], as they comment: ‘The deterministic perfect soliton crystal states are believed to be caused by either the pump-SBL pair induced cross-phase modulation [35] or two co-lasing lasers due to the insufficient filtering effect of the BPF during the turnkey process’. It must be stated, however, that this is a phase phenomenon (‘pump-SBL pair induced cross-phase modulation’). I would guess that it will depend on the relative frequency between the pair. Can the authors comment on that and on their ability to control different types of states? Do they have a specific range of parameters where they can achieve deterministic generation?

Response: In short, the soliton existing range is non-degenerate due to the compensated thermal nonlinearity by the two coexisting lasers in the microresonator. Therefore, different SBL detunings

results in different soliton states. The MRT can change the SBS frequency shift, and consequently the SBL detuning and the soliton number. We have developed a comprehensive model and reproduced turnkey soliton in the revised Supplementary Information Section III. There, we show that the MRT range for stable SBL soliton generation is ~ 0.8 K. By changing MRT, we can achieve deterministic soliton number selection from 1 to 3. The temperature step is around 0.3 K, which depends on the temperature sensitivity of the SBS frequency shift.

Besides perfect soliton crystal (PSC) states, we did not observe any other multi-soliton states such as soliton molecules or defects in the experiment. We attribute the dominant PSC existence over other multi-soliton states to the equally spaced potential well [44] created by co-lasing pump modes due to the insufficient BPF out-of-band suppression. We have revised the description in the Results section to make it clear and included a discussion in the revised Supplementary Information Section IV.

Revisions made: We have added the explanation for the deterministic selection of soliton number in the revised Supplementary Information Section II.

“Figure S4 shows the soliton number dynamics when changing the microresonator temperature in the soliton existing regime. The ECDL is swept from blue side to red side. By changing the microresonator temperature, the SBS frequency shift is changed thus both the SBL detuning and thermal equilibrium condition are changed when pump detuning is near zero. With two intracavity lasers compensated thermal effect, the SBL soliton existing regime is no longer degenerate due to the XPM effect [4–7]. The soliton steps showing the soliton number switching are clearly observed in Fig. S4. By changing the temperature, we can deterministically select the soliton number from 3 to 1 before pump power drops or pump laser exits the microresonator at what time the pump detuning is around zero detuning. Therefore, in our microresonator-filtered fiber laser, we can deterministically select the soliton number by changing the microresonator temperature.”

Fig. S4. Soliton dynamics by sweeping the pump laser frequency under different microresonator temperatures. (a) the final soliton number (before pump power drops or pump laser exits the microresonator) is 3; (b) the final soliton number is 2; (c) the final soliton number is 1.”

We have revised the description of perfect soliton crystal (PSC) states in the Results section. “Besides PSC states, we did not observe any other multi-soliton states such as soliton molecules or defects in the experiment. We attribute the dominant PSC existence over other multi-soliton states to the equally spaced potential well [44] created by co-lasing pump modes due to the insufficient BPF out-of-band suppression (see Supplementary Information Section IV).”

We have also included a discussion about the PSC generation in the revised Supplementary Information Section IV.

“We attribute the turnkey perfect soliton crystal (PSC) generation in our microresonator-filtered laser to the equally spaced potential well [8] created by co-lasing pump mode lasers due to the insufficient filtering effect of the BPF. Figure S8 shows the spectrum of the pump mode at the output of the EDFA when SBL soliton is generated. Besides the main lasing signal, there are three other lasing signals for the pump mode. The co-lasing signals will create equally spaced potential well ranging from 1 FSR to 3 FSRs through XPM effect, attract soliton to be equally spaced in time domain and lead to PSC. Of note, pump mode FSR is almost the same with the Brillouin mode FSR in our FP microresonator made of graded-index fiber.

Fig. S8. Pump mode spectrum. One of the pump mode lasers is buried due to the limited resolution of the optical spectrum analyzer (YOKOGAWA, AQ6370D), as indicated by the red arrow. The Brillouin mode suppression is larger than 40 dB so that SBL comb lines can not be observed clearly.”

Reviewer #4 (Remarks to the Author):

Authors of the paper entitled "Turnkey photonic flywheel in a Chimera cavity" present a chimera cavity within which they integrate a compact fiber Fabry-Perot. The fiber-based gain medium within the chimera cavity supplies energy for a Brillouin laser that then acts as a pump for a Kerr comb. This work builds on their prior demonstration (PRL 2020 seen as Ref. 8) which explores external pumping of a Brillouin laser-driven Kerr comb as a means of obtaining improved phase noise and better frequency stability. The primary mechanisms for improved performance are (1) the pump frequency noise suppression supplied by the internal Brillouin laser dynamics and (2) a reduction in photothermal noise produced competing effects within the resonators. The results are impressive and the relatively repeatable self-starting of the stable comb generation within this chimera cavity appears to be a substantive advance toward the turn-key operation of such novel laser systems. For these reasons, I believe that this work has the potential for high impact and will be of significant interest to the broad readership of Nat. Comm. For these reasons, I support the publication of this work in Nat. Comm. However, I do have a few points of clarification that I think the authors should address prior to publication below.

Response: We thank Reviewer #4 for well receiving the key points and recognizing the importance of our work

Comments:

- The following statement was unclear "Of note, our deterministic turnkey process is independent of 980-nm laser power of the EDFA in an offset range of ± 100 mW, where the reported single LCS attractor can be destroyed [30]." In particular, it was unclear to me what is meant by "an offset range of ± 100 mW" in this context. Please clarify this point.

Response: We thank Reviewer #4 for the careful reading. What we want to express here is that our turnkey soliton has a large tolerance of ± 100 mW with respect to the 980-nm laser power of the EDFA. For example, with 1.0-W 980-nm laser power, we can obtain the turnkey 2-soliton state. By increasing or decreasing the 980-nm laser power to 1.1 W or 0.9 W, we can still get the turnkey soliton at the same state.

Revisions made: We have clarified in the revised manuscript in Section Discussion as "For example, the pump power tolerance of our deterministic turnkey process is ± 100 mW, an order of magnitude larger than the previous self-emergent LCS [53]."

- The authors identify a number of applications that this system could enable. Many of these applications rely heavily on the production of stable, low phase noise frequency combs. Through their measurements the authors quantify the phase noise at a 1 MHz offset frequency. However, the close to carrier (< 1 MHz) phase noise is also very important for these applications. I think it would also be good to discuss the phase noise at lower offset frequencies and perhaps discuss the mechanisms that contribute to this reduced phase noise.

Response: Thanks for the suggestion. The reviewer might have overlooked the description in the original manuscript where we pointed out the phase noise at low offset frequencies as “The measured SSB phase noises at 10 kHz, 100 kHz, and 1 MHz offset frequencies are -128 dBc/Hz, -147 dBc/Hz, and -166 dBc/Hz, respectively.” and “following $1/f^2$ trend with the offset frequencies”.

Revisions made: We have conducted comprehensive analysis on the mechanisms for noise reduction in the revised Supplementary Information Section VIII.

“VIII Theory for SBL soliton noise suppression

SBL and comb linewidth narrowing When the Brillouin mode resonance peak match the SBS gain peak, the SBL narrowing factor r is determined by $r = (1 + \Gamma_b/\gamma)^2$ [9–11], where Γ_b is the Brillouin gain bandwidth and γ is the Brillouin mode cavity linewidth. Therefore, larger Q factor can lead to better SBL noise suppression and larger linewidth narrowing factor. Since Γ_b is usually larger than 10 MHz and γ is smaller than 1 MHz for our MMF microresonator, a linewidth narrowing factor of more than 100 (20 dB) is expected, as verified by our previous experimental results [12].

For SBL soliton, the fundamental comb linewidth is the same with the SBL [13]. Despite the compromised SBL linewidth narrowing factor by 3-5 times from the detuning effect [14], the SBL comb linewidth narrowing effect from the pump can still be larger than 20 dB.

SBL soliton jitter reduction The detuning is key to soliton’s performance, including the soliton pulse width, soliton energy and so on. The detuning instability will cause the soliton jitter through fluctuation of Kerr nonlinearity from the affected soliton’s pulse width and energy. Therefore the detuning noise suppression can lead to soliton jitter reduction.

According to the theoretical analysis [15], the relationship between the SBL detuning σ_B and pump detuning σ_F is described as

$$\sigma_B = \frac{\sigma_F - [(\Omega_b^2 - \Omega^2)/2\Omega]}{1 + \Gamma_b/\gamma} \quad (1)$$

where Ω_b is the coustic frequency at SBS gain peak and Ω is the actual SBS frequency shift. Therefore, the SBL detuning noise can be largely suppressed from $d\sigma_B = d\sigma_F/(1 + \Gamma_b/\gamma)$. Besides, larger Q factor can lead to larger SBL detuning noise suppression thus the SBL soliton jitter reduction.”

- To provide context, please make a brief comparison of the noise characteristics obtained with this system and compare with Kerr combs systems of similar size-scale.
- Also, a brief comparison to their earlier work in PRL seen as reference 8 could be instructive.

Revisions made: In the revised Supplementary Information Section X, we have conducted comprehensive comparison with the reported results, in terms of both the fundamental comb linewidth and soliton jitter.

“X Noise performance comparison

Fundamental comb linewidth Table 1 lists the fundamental comb linewidth in different platforms and with different methods. According to Table 1, in general, our SBL soliton comb linewidth shows the best result among the reported results.

Table 1. Comparison of DKS microcomb linewidth

Material	Q factor (Million)	Microcomb linewidth (Hz)	Soliton active control	Reference
SiO ₂	278	~0.1	W/o (microresonator-filtered laser)	This work
SiO ₂	384	~0.4	W/o (SBS)	[12]
SiO ₂	44.4	24	W/o (SBS)	[21]
Si ₃ N ₄	56	~10	W/o (SIL*)	[17]
Si ₃ N ₄	11.6	~1000	W/	[22]

*SIL: self-injection locking.

For SBL soliton, the fundamental comb linewidth is the same with the SBL [13]. Based on the analysis in Section VII, the SBL linewidth narrowing factor depends on both the Q factors and SBS gain bandwidth. Therefore, larger Q factors and larger SBS gain bandwidth can lead to lower SBL and its comb fundamental linewidth. For example, in this work with microresonator-filtered laser and our previous SBL soliton comb pumped by an ECDL [12], the MMF microresonator has larger Q factors of ~300 million, resulting in sub-hertz fundamental comb linewidth. However, for the silica disk microresonator [21], the Q factor is 44.4 million, which is 10 times lower than our MMF microresonator and this results in 24-Hz comb fundamental linewidth due to the smaller linewidth narrowing factor.

For the soliton achieved by the self-injection locking method, the comb fundamental linewidth is usually determined by the Q-factor [17] and is eventually limited by the small mode volume induced large thermo-refractive noise for the on-chip microresonators [17]. For example, for the on-chip Si₃N₄ microresonator with high Q factor of 56 million, the fundamental comb linewidth is ~10 Hz which is limited by the thermo-refractive noise.

The pump laser phase noise suppression in our microresonator-filtered laser can also be viewed as a result of self-injection locking, which is discussed in detail in Section VIII. The ultrahigh Q factor of ~300 million is responsible for the 0.1-Hz fundamental linewidth for the pump laser. Besides, the pump RIN eventually limits the pump laser fundamental linewidth to be lower than 0.1 Hz.

Comb repetition rate Table 2 lists the state-of-the-art phase noise of DKS microcomb repetition rate in various platforms with different methods. According to Table 2, in general, our SBL soliton jitter (either in microresonator-filtered laser or externally pumped) in MMF microresonator, outperforms all other DKS microcombs, with or without active control, except the one study in a MgF₂ crystalline microresonator [23] where a sideband Pound–Drever–Hall locking was implemented to optimize and stabilize the detuning setpoint via the ultrastable laser for the quiet point (QP) operation that requires either avoided mode crossing or large third order dispersion. Our approach based on SBS process in MMF microresonator, on the other hand, is completely free running without the need of any active control.

Table 2. Comparison of phase noise of DKS microcomb repetition rate

Material	Configuration	Carrier frequency (GHz)	SSB phase noise (dBc/Hz, scaled to 10 GHz)		Soliton active control	Reference
			10 kHz	100 kHz		
SiO ₂	Bright soliton	10.08/20.16	-128	-147	W/o (microresonator-filtered laser)	This work
SiO ₂	Bright soliton	10.08/20.16	-125	-148	W/o (SBS)	[12]
SiO ₂	Bright soliton	0.945	-120	-140	W/o (SBS)	[11]
SiO ₂	Bright soliton	10.43	-125	-144	W/o (SBS)	[21]
SiO ₂	Bright soliton	11.02	-120	-139	W/o (SBS)	[24]
Si ₃ N ₄	Dark soliton	5.4	-108	-134	W/o (SIL)	[17]
MgF ₂	Bright soliton	9.9	-130	-130	W/o (SIL)	[25]

SiO ₂	Bright soliton	22	-111	-147	W/ (QP)	[19]
SiO ₂	Bright soliton	15.2	-117	-143	W/ (QP)	[26]
SiO ₂	Bright soliton	11.4	-134	-143	W/ (QP)	[27]
MgF ₂	Bright soliton	14.09	-142	-159	W/ (QP)	[23]
Si ₃ N ₄	Bright soliton	9.78	-110	-130	W/	[28]

*SIL: self-injection locking.

Our previous SBL soliton jitter in a monolithic fiber resonator based on highly nonlinear fiber reaches the quantum noise limit [11]. Here the MMF microresonator in this work has two-fold advantages, the large mode volume and lower Kerr nonlinearity parameter, which can efficiently lower the quantum noise limit [20]. Therefore, the SBL soliton jitter in this work and our recent work [12] is not limited by the quantum noise, but the SBL RIN, as analyzed in Section IX. Besides, according to the detuning noise suppression analysis in Section VII, higher Q factor will lead to better detuning noise suppression thus lower soliton jitter. This should be the reason why our soliton jitter results with ultrahigh-Q MMF microresonator outperforms the other SBL soliton jitter results even though they are not limited by the quantum noise.”

- To build on this work, I think it would be invaluable to include estimates and a discussion of noise sources and their contributions to the dynamics of this system.

Revisions made: We comprehensively discuss the noise sources and their contributions to the dynamics of this system, which are included in the revised Supplementary Information Section IX.

“IX Analysis on the phase noise of the generated microwave signal

SBS frequency shift According to the frequency noise measurement in Fig. 5a and Fig. S3a, the laser phase noise PSDs of both pump and SBL follow $1/f^2$ trend with the offset frequencies and are dominated by the pump RIN as shown in Section VIII. Therefore, the phase noise PSD of the SBS frequency shift sees noise suppression of 10 dB/decade from the pump or SBL phase noise and follows $1/f^2$ trend with the offset frequencies as shown in Fig. 5c. The phase noise PSD with $1/f^2$ trend is mainly dominated by frequency flicker noise resulting from pump laser frequency drift. Despite the common mode noise suppression of 10 dB/decade, the coherence between pump and SBL is still worse compared to the one between phase locked SBL comb lines. Of note, for the external pumped case [12], the phase noise PSD of the SBS frequency shift is dominated by the pump laser phase noise and no noise suppression with respect to the pump laser is observed [12].

SBL soliton comb repetition rate As demonstrated in Section VII, the soliton detuning noise is largely suppressed due to the SBS process, leading to reduced timing jitter. We measure the SSB phase noise of the soliton repetition rate with an all-fiber reference-free Michelson interferometer (ARMI) timing jitter measurement apparatus. The error signal is fed back to the PZT stretcher in the fiber link, and the timing delay provided by the long fiber link is locked to the free-running comb. As a result, the timing jitter PSD of the free-running comb can be measured outside the locking bandwidth [18,19]. It is important to keep the locking bandwidth as low as possible to ensure broadband characterization. In our work, we used a 100 Hz locking bandwidth so that the SSB phase noise can be precisely measured from 100 Hz to higher offset frequencies.

As shown Fig. 5c, the phase noise PSD of the comb repetition rate follows $1/f^2$ trend with the offset frequencies, which means white frequency noise of the generated microwave signal. However, the phase noise PSD is still 15 dB larger than the calculated quantum noise limit (Fig. S12) [12,20]. According to our previous analysis [12], we attribute that the phase noise of the SBL soliton comb repetition rate is dominated by the SBL RIN and pump RIN, which depends on the 980-nm laser RIN as analyzed in Section II. In particular, for the offset frequencies below 1 kHz, the phase noise of the comb repetition rate is dominated by the technical noise from the EDFA whose gain response time is around ms level. Of note, the SSB phase noise below 10 kHz can be easily reduced by a feedback loop.

Fig. S12. Quantum noise limit for our MMF microresonator.

In terms of long-term stability, the key is the pump laser frequency which can be affected by the large fiber cavity length, the temperature of the microresonator, the gain stability in the EDFA and so on. One of the examples is in Fig. 4 we show the dynamics of comb power and repetition rate when we modulate the large fiber cavity length. Therefore, we can improve the long-term stability by shielding the whole system, using high-precision temperature controller, introducing feedback on the 980-nm laser and so on.”

For this and future works I think it would be incredibly valuable to develop models that include photothermal effects brew on dynamics, nonlinearities, etc. Certainly to include all of these would substantially add to the length of the paper, and may not be reasonable. Nevertheless, progress in this direction I think would be invaluable to understanding the true potential of such hybrid laser systems.

Revisions made: We have developed a comprehensive model and reproduced turnkey soliton in the revised Supplementary Information Section III. We have added the related description in the main text as “We also numerically reproduce the turnkey soliton and conduct comprehensive analysis in Supplemental Information Section III.”

REVIEWERS' COMMENTS

Reviewer #2 (Remarks to the Author):

The authors properly addressed all my concerns, and I think the manuscript is ready to be published.

Reviewer #4 (Remarks to the Author):

The authors have made a very thorough and rigorous response to all of my queries. They have also revised the manuscript to improve clarity. I am satisfied with all of the changes and am in favor of publication.